# Diffusion Bridge or Flow Matching?
# A Unifying Framework and Comparative Analysis

**Kaizhen Zhu** [1 2 *]   **Mokai Pan** [1 2 *]   **Zhechuan Yu** [1]   **Jingya Wang** [1]   **Jingyi Yu** [1]   **Ye Shi** [1 2 †]

## Abstract

Diffusion Bridge and Flow Matching have both demonstrated compelling empirical performance in transformation between arbitrary distributions. However, there remains confusion about which approach is generally preferable, and the substantial discrepancies in their modeling assumptions and practical implementations have hindered a unified theoretical account of their relative merits. We have, for the first time, provided a unified theoretical and experimental validation of these two models. We recast their frameworks through the lens of Stochastic Optimal Control and prove that the cost function of the Diffusion Bridge is lower, guiding the system toward more stable and natural trajectories. Simultaneously, from the perspective of Optimal Transport, interpolation coefficients $t$ and $1 - t$ of Flow Matching become increasingly ineffective when the training data size is reduced. To corroborate these theoretical claims, we propose a novel, powerful architecture for Diffusion Bridge built on a latent Transformer, and implement a Flow Matching model with the same structure to enable a fair performance comparison in various experiments. Comprehensive experiments are conducted across Image Restoration, Translation, and Style Transfer tasks, systematically varying both the distributional discrepancy (different difficulty) and the training data size. Extensive empirical results align perfectly with our theoretical predictions and allow us to delineate the respective advantages and disadvantages of these two models. Our code is available at https://github.com/zhukaizhen/diffusion_bridge_flow_matching.

*Equal contribution [1]ShanghaiTech University [2]InstAdapt. Correspondence to: Ye Shi <shiye@shanghaitech.edu.cn>.

*Proceedings of the $43^{rd}$ International Conference on Machine Learning*, Seoul, South Korea. PMLR 306, 2026. Copyright 2026 by the author(s).

## 1. Introduction

Diffusion models have been widely used in a variety of applications, demonstrating remarkable capabilities and promising results in numerous tasks such as image generation (Ho et al., 2020; Song et al., 2020; Xia et al., 2023), video generation (Ho et al., 2022; Luo et al., 2023a), imitation learning (Chi et al., 2023; Ze et al., 2024; Wu et al., 2024), and reinforcement learning (Ding et al., 2024; 2026; Wang et al., 2022; Ada et al., 2024), etc. However, standard diffusion models exhibit inherent limitations that they are difficult to achieve the conversion between any two distributions since its prior distribution is assumed to be Gaussian noise. They can also rely on meticulously designed conditioning mechanisms and classifier/loss guidance (Chung et al., 2023; Yang et al., 2024; Kawar et al., 2022) to facilitate conditional sampling and ensure output alignment with a target distribution. However, these methods can be cumbersome and hard to guarantee the diffusion can finally reach the desired conditional distribution.

To address this challenge, Diffusion Bridge and Flow Matching are two prevalent approaches for achieving distribution-to-distribution transformation. As for diffusion bridges, Schrödinger Bridge (Liu et al., 2023; Shi et al., 2024; De Bortoli et al., 2021) deterministically steers one prescribed probability measure to another via the minimum-entropy re-weighting of an underlying reference stochastic process, producing a path-wise coupling whose time-marginals coincide with the specified endpoint distributions. DDBMs (Zhou et al., 2023), GOUB (Yue et al., 2024), and related methods enforce end-to-end exact matching by incorporating Doob's $h$-transform into the forward Stochastic Differential Equations (SDE) of the diffusion process. UniDB (Zhu et al., 2025) further reformulates and unifies the diffusion bridge paradigm through a stochastic optimal control framework.

Flow-based generative models have undergone rapid evolution, progressing from the early invertible transformations of Normalizing Flows (Papamakarios et al., 2021; Albergo & Vanden-Eijnden, 2022; Mathieu & Nickel, 2020) to the recently emerged paradigm of Flow Matching. Continuous Normalizing Flows rely on repeated calls to high-order ODE solvers during forward-backward propagation, resulting in

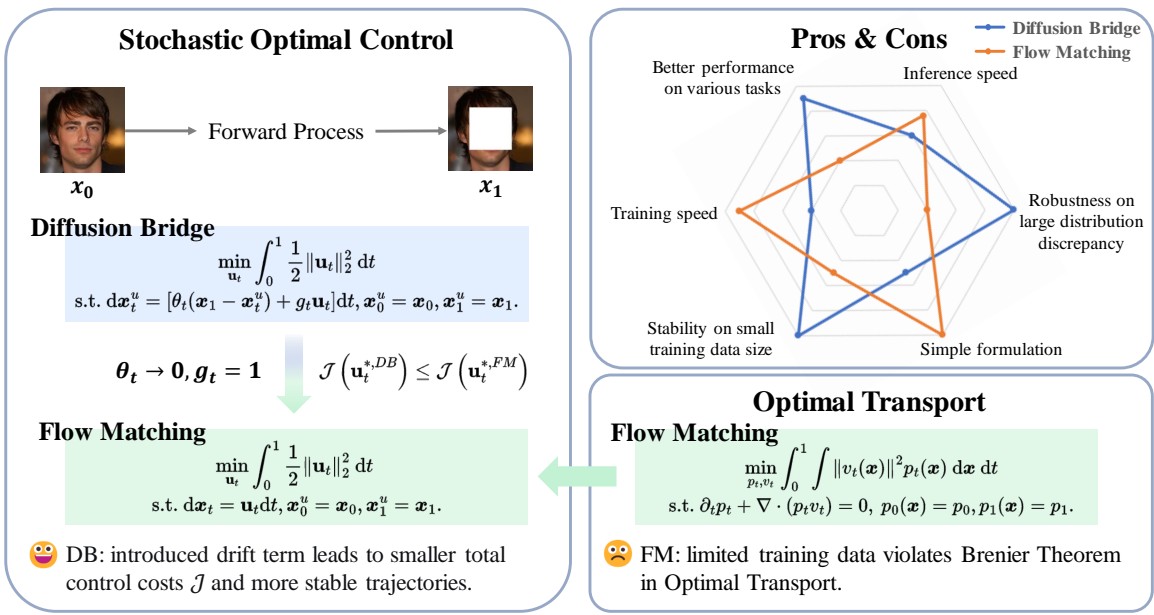

Figure 1. Overview of Diffusion Bridge (DB) versus Flow Matching (FM) on mapping one distribution to another. From the perspective of Stochastic Optimal Control and Optimal Transport, we demonstrate that 1) the cost function of DB is lower than that of FM, implying a more stable and natural trajectory of DB; 2) the linear interpolation scheme in FM (with coefficients $t$ and $1-t$) becomes suboptimal under limited training data regimes. The respective strengths and weaknesses of DB and FM are summarized in the radar chart.

a training procedure dominated by costly simulation-and-differentiation loops that severely limit scalability. Flow Matching (Lipman et al., 2023; 2024; Liu et al., 2022) circumvents this bottleneck by first designing a probability path that interpolates between a prior and the data distribution, and then directly learning a time-dependent vector field whose integral trajectories realize this path. Its simple modeling principle and high generation quality have attracted significant attention. OT-CFM (Tong et al., 2023) further stabilizes training and improves sample fidelity by performing optimal-transport pairing of noise and data samples before learning the vector field.

Diffusion Bridges and Flow Matching successfully address a wide spectrum of tasks—including image restoration (Luo et al., 2023b; Yue et al., 2024; Zhu et al., 2025; Martin et al., 2025; Cohen et al., 2025), translation (Zhou et al., 2023; Zheng et al., 2024), text-to-image generation (Liu et al., 2025; Xu et al., 2025), and robotic policy synthesis (Zhang et al., 2025b)—yet a comprehensive theoretical framework that rigorously delineates their mutual relationship, comparative advantages, and inherent limitations remains open. This naturally leads to a fundamental question:

> *"Mapping one distribution to another, which is better—Diffusion Bridge or Flow Matching?"*

In this paper, we firstly theoretically analyze the relationship between Diffusion Bridge and Flow Matching, and experimentally compare their respective advantages and disadvantages in implementing distribution-to-distribution transformation.

**Comparative Theoretical Framework for Diffusion Bridge and Flow Matching.** From the perspective of Stochastic Optimal Control, we provide a unified framework for Diffusion Bridge and Flow Matching, demonstrating that the cost function of Diffusion Bridge is lower than that of Flow Matching, implying a more stable and natural trajectory of Diffusion Bridge. Furthermore, we analyze the linear interpolation scheme of Flow Matching (governed by coefficients $t$ and $1-t$) from an Optimal Transport perspective. Our analysis reveals that this scheme can lead to performance degradation, particularly when the training dataset size is limited.

**Comparative Experimental Evaluation of Diffusion Bridge versus Flow Matching.** To enable a fair and controlled empirical evaluation, we design a new, powerful neural architecture for Diffusion Bridge models based on a latent Transformer. This architectural innovation significantly enhances the capability of DB models. Using this and an equivalent-structure Flow Matching model, we conduct extensive experiments across a diverse suite of tasks, including Image Inpainting (under varying training data sizes and distributional discrepancies), Super-Resolution, Deblurring, and Translation tasks. This systematic evaluation clearly shows the respective strengths and weaknesses of both methodologies in Figure 1.

## 2. Related Work

**Diffusion Bridge.** Diffusion Bridge models (Li et al., 2023; Lyu et al., 2024; Kim et al., 2024; Zhang et al., 2025a; Li et al., 2026; Sobieski et al., 2025; Kieu et al., 2025; Wang et al., 2025; Luo et al., 2025b) are relatively advanced methods for achieving transformation between distributions. Diffusion Schrödinger Bridges (De Bortoli et al., 2021; Chen et al., 2021; Liu et al., 2023; Shi et al., 2024; Deng et al., 2024) construct a transport mapping between $p_{\text{data}}$ and $p_{\text{prior}}$ by minimizing the KL divergence $\pi^*$, using IPF to alternately optimize boundary conditions. However, it suffers from high computational complexity, especially in high dimensions, making direct optimization challenging. DDBMs (Zhou et al., 2023) and GOUB (Yue et al., 2024) achieve promising results by incorporating Doob's $h$-transform into the forward SDE of diffusion models to pin the terminal distribution to a specific target. While DDBMs establish that Flow Matching is a special case of DDBMs-VE in its zero-noise limit, a systematic comparison of how different diffusion bridge and flow matching formulations affect generative performance remains unexplored. UniDB (Zhu et al., 2025) reformulates the Diffusion Bridge as a Stochastic Optimal Control problem, proving that Doob's $h$-transform is a special case when the terminal penalty coefficient $\gamma \to \infty$, thereby unifying and generalizing existing diffusion bridges.

**Flow Matching.** Flow-based generative models (Klein et al., 2023; Eijkelboom et al., 2024; Ohayon et al., 2024; Jing et al., 2024; Guo et al., 2024; Braun et al., 2024; Hu et al., 2024b; Ding et al., 2025; Luo et al., 2025a; Geng et al., 2026; Kulikov et al., 2025) have evolved from Continuous Normalizing Flows (CNFs) (Papamakarios et al., 2021; Albergo & Vanden-Eijnden, 2022; Mathieu & Nickel, 2020), which treat data generation as an invertible, continuous-time transformation governed by an ordinary differential equation (ODE). CNFs admit exact likelihood evaluation and an invertible mapping, but their integration cost scales with dimension and integration steps, making training and sampling expensive. To overcome this computational bottleneck, Flow Matching (Lipman et al., 2023; 2024; Liu et al., 2022) trains the model by regressing onto the conditional optimal vector field that maps distributions to distributions paired with their conditions; the resulting conditional vector field can be computed in closed form without simulating the ODE. While the conditional path is a straight-line interpolation in Euclidean space. Optimal Transport Conditional Flow Matching (OT-CFM) (Tong et al., 2023) replaces the independent coupling with a static optimal-transport plan, yielding conditional trajectories that are globally straighter and reduce curvature. This mitigates the back-tracking phenomenon observed in diffusion-like schedules and accelerates both training and sampling.

**Stochastic Optimal Control.** The incorporation of SOC

principles into diffusion bridges and flow matching models has emerged as a promising paradigm for effectively guiding distribution transitions. I2SB (Liu et al., 2023) firstly introduced the SOC formulation for modeling diffusion bridges. RB-Modulation (Rout et al., 2024), NDTM (Pandey et al., 2025), and HardFlow (Li et al., 2025) introduced SOC via a simplified SDE structure for training-free style transfer and solving inverse problems based on pre-trained diffusion models. DBFS (Park et al., 2024) leveraged SOC to construct diffusion bridges in infinite-dimensional function spaces and established equivalence between SOC and Doob's $h$-transform. Adjoint Matching (Domingo-Enrich et al., 2025) for the first time, systematically unifies Flow Matching with SOC, and on this basis proposes a novel, theoretically unbiased reward fine-tuning framework. (Mei et al., 2025) proposes a Flow-Matching-based framework for SOC, which reformulates the classical SOC problem into a data-driven optimization task, thereby circumventing the intractability of solving high-dimensional nonlinear Hamilton-Jacobi-Bellman (HJB) equation.

## 3. Preliminaries

### 3.1. Stochastic Optimal Control

Stochastic Optimal Control (SOC) offers a principled methodology for designing optimal policies in dynamical systems operating under uncertainty, with applications spanning multiple disciplines (Geering et al., 2010; Chen et al., 2023; Rout et al., 2024; Zhu et al., 2025; Pandey et al., 2025; Li et al., 2025). We consider the following SOC formulation with quadratic costs (Kappen, 2008):

$$\min_{\mathbf{u}_{t,\gamma}} \mathbb{E}\left[\int_0^T \frac{1}{2}\|\mathbf{u}_{t,\gamma}\|_2^2\, dt + \frac{\gamma}{2}\|\boldsymbol{x}_T^u - \boldsymbol{x}_T\|_2^2\right] \quad (1)$$

s.t. $\mathrm{d}\boldsymbol{x}_t^u = [\mathbf{f}(\boldsymbol{x}_t^u, t) + g_t\mathbf{u}_{t,\gamma}]\mathrm{d}t + g_t\mathrm{d}\boldsymbol{w}_t,\ \boldsymbol{x}_0^u = \boldsymbol{x}_0,$

where $\boldsymbol{x}_t^u$ is the diffusion process under control, $\boldsymbol{x}_0$ and $\boldsymbol{x}_T$ denote the predetermined initial and terminal states, respectively, $\|\mathbf{u}_{t,\gamma}\|_2^2$ is transient cost, and $\frac{\gamma}{2}\|\boldsymbol{x}_T^u - \boldsymbol{x}_T\|_2^2$ corresponds to the terminal cost with its penalty coefficient $\gamma$. The expectation value is over all stochastic trajectories originating from $\boldsymbol{x}_0$ (Kappen, 2008). The whole SOC problem (1) aims to design the controller $\mathbf{u}_{t,\gamma}$ to drive the dynamics from $\boldsymbol{x}_0$ to $\boldsymbol{x}_T$ while minimizing the overall cost.

### 3.2. Flow Matching

Notably, most Flow Matching (FM) work (Lipman et al., 2023; 2024; Liu et al., 2022; Albergo & Vanden-Eijnden, 2022; Zhang et al., 2025b; Xing et al., 2025; Yao et al., 2025; Yi et al., 2025; Cao et al., 2025) directly models the generative process, which is different from standard diffusion and diffusion bridges (Ho et al., 2020; Song et al., 2020; Zhou et al., 2023). To avoid discrepancy, we decouple FM with

both forward and sampling processes (Liu et al., 2022) in the context of diffusion models. The FM forward process is to transform samples $\boldsymbol{x}_0 \in \mathbb{R}^d$ from a source (data) distribution $p_0$ into $\boldsymbol{x}_1 \in \mathbb{R}^d$ from a target (prior) distribution $p_1$ by defining a flow $\psi_t : [0, 1] \times \mathbb{R}^d \to \mathbb{R}^d$ parameterized by a learnable vector field $v_t(\boldsymbol{x})$. Due to the intractability of the true vector field in practice, the conditional flow $\psi_t(\boldsymbol{x} \mid \boldsymbol{x}_1)$ is constructed under the probability paths $p_t(\boldsymbol{x} \mid \boldsymbol{x}_1)$ with the vector field $v_t(\boldsymbol{x} \mid \boldsymbol{x}_1)$ (Lipman et al., 2023). The conditional flow matching training objective is formed as

$$\mathcal{L}_{\text{FM}}(\theta) = \mathbb{E}_{t, \boldsymbol{x}_0, \boldsymbol{x}_1, \boldsymbol{x}_t} \left[ \left\| \boldsymbol{v}_\theta \left( \boldsymbol{x}_t, t \right) - \left( \boldsymbol{x}_1 - \boldsymbol{x}_0 \right) \right\|^2 \right]. \quad (2)$$

To find a useful conditional flow $\psi_t(\boldsymbol{x} \mid \boldsymbol{x}_1)$, one popular example is the minimizer of the dynamic Optimal Transport (OT) problem with quadratic cost (Villani et al., 2008; Villani, 2021; Peyré & Cuturi, 2019), which is formalized as

$$\min_{p_t, v_t} \int_0^1 \int \left\| v_t(\boldsymbol{x}) \right\|^2 p_t(\boldsymbol{x}) \mathrm{d}\boldsymbol{x} \mathrm{d}t \quad (3)$$
$$\text{s.t. } \partial_t p_t + \nabla \cdot (p_t v_t) = 0, p_0(\boldsymbol{x}) = p_0, p_1(\boldsymbol{x}) = p_1,$$

where the objective is the Wasserstein-2 transport cost between two distributions, and the conditions are that the vector field $v_t$ satisfies the continuity equation and the initial and terminal conditions. The solution $(p_t^*, v_t^*)$ defines a flow called an OT displacement interpolant as $\psi_t^*(x) = tT(x) + (1 - t)x$ where $T : \mathbb{R}^d \to \mathbb{R}^d$ is the OT map such that $T_\# p_0 = p_1$ (i.e., $p_1$ is the push-forward of $p_0$ under the transport map $T$). By defining the random variable $\boldsymbol{x}_t = \psi_t^*(\boldsymbol{x}_0)$ with $\boldsymbol{x}_0 \sim p_0$ and finding a bound for the Wasserstein-2 transport cost (Liu et al., 2022; Lipman et al., 2024), the dynamic OT problem leads to the following variational problem for $\eta_t = \psi_t(\boldsymbol{x}_0 \mid \boldsymbol{x}_1)$:

$$\min_{\eta_t} \int_0^1 \left\| \dot{\eta}_t \right\|^2 \mathrm{d}t \quad \text{s.t. } \eta_0 = \boldsymbol{x}_0, \eta_1 = \boldsymbol{x}_1. \quad (4)$$

According to the Euler-Lagrange equations (Gelfand et al., 2000), the minimizer can be obtained as

$$\boldsymbol{x}_t = \psi_t^*(\boldsymbol{x}_0 \mid \boldsymbol{x}_1) = \eta_t^* = t\boldsymbol{x}_1 + (1 - t)\boldsymbol{x}_0. \quad (5)$$

For backward sampling, after we get $\hat{\boldsymbol{v}}_\theta$, we solve the ODE $\mathrm{d}\tilde{\boldsymbol{x}}_t = -\hat{\boldsymbol{v}}_\theta \mathrm{d}t$ starting from $\tilde{\boldsymbol{x}}_0 \sim p_1$ to transfer $p_1$ to $p_0$ and set $\tilde{\boldsymbol{x}}_t = \boldsymbol{x}_{1-t}$ according to the time-symmetric property (Liu et al., 2022). For more details, please refer to (Lipman et al., 2024).

### 3.3. Diffusion Bridges

$h$-transform-based diffusion bridges, such as DDBMs (Zhou et al., 2023) and GOUB (Yue et al., 2024), achieve promising results by modifying the original forward SDE to

pass through the predetermined terminal. UniDB further constructs diffusion bridges through the Stochastic Optimal Control (SOC) problem and generalizes these $h$-transform-based methods. Based on the Generalized Ornstein-Uhlenbeck (GOU) process (Ahmad, 1988; Yue et al., 2024),

$$\mathrm{d}\boldsymbol{x}_t = \theta_t \left( \boldsymbol{\mu} - \boldsymbol{x}_t \right) \mathrm{d}t + g_t \mathrm{d}\boldsymbol{w}_t, \quad (6)$$

where $\boldsymbol{\mu}$ is a given state vector, $\boldsymbol{w}_t$ is the Wiener process, $\theta_t$ and $g_t$ denote the scalar drift and diffusion coefficient with the relationship $g_t^2 = 2\lambda^2 \theta_t$ where $\lambda^2$ is a given positive constant scalar, a specific example of the UniDB framework is introduced as UniDB-GOU (Zhu et al., 2025) and the related SOC problem constructs the forward bridge process as:

$$\min_{\mathbf{u}_{t,\gamma}} \mathbb{E} \left[ \int_0^1 \frac{1}{2} \| \mathbf{u}_{t,\gamma} \|_2^2 \mathrm{d}t + \frac{\gamma}{2} \| \boldsymbol{x}_1^u - \boldsymbol{x}_1 \|_2^2 \right]$$
$$\text{s.t. } \mathrm{d}\boldsymbol{x}_t^u = \left[ \theta_t(\boldsymbol{x}_1 - \boldsymbol{x}_t^u) + g_t \mathbf{u}_{t,\gamma} \right] \mathrm{d}t + g_t \mathrm{d}\boldsymbol{w}_t, \ \boldsymbol{x}_0^u = \boldsymbol{x}_0, \quad (7)$$

where $\boldsymbol{\mu} = \boldsymbol{x}_1$ in the SDE is set as the final condition. To be consistent with flow matching above, here we change the original time schedule $t \in [0, T]$ to $t \in [0, 1]$. According to the certainty equivalence principle (Chen et al., 2023; Rout et al., 2024), UniDB derives the same optimal controller $\mathbf{u}_{t,\gamma}^*$ by modifying the SOC problem (7) to one with the deterministic ODE condition as follows, specifically,

$$\min_{\mathbf{u}_{t,\gamma}} \int_0^1 \frac{1}{2} \| \mathbf{u}_{t,\gamma} \|_2^2 \mathrm{d}t + \frac{\gamma}{2} \| \boldsymbol{x}_1^u - \boldsymbol{x}_1 \|_2^2$$
$$\text{s.t. } \mathrm{d}\boldsymbol{x}_t^u = \left[ \theta_t(\boldsymbol{x}_1 - \boldsymbol{x}_t^u) + g_t \mathbf{u}_{t,\gamma} \right] \mathrm{d}t, \ \boldsymbol{x}_0^u = \boldsymbol{x}_0. \quad (8)$$

Previous works like GOUB (Yue et al., 2024) can be considered as taking $\gamma \to \infty$ in (8), which means the controlled dynamics would pass precisely through the preset terminal $\boldsymbol{x}_1$ (Chen et al., 2023), therefore, we can transform the SOC problem (8) with $\gamma \to \infty$ into the following form:

$$\min_{\mathbf{u}_t} \int_0^1 \frac{1}{2} \| \mathbf{u}_t \|_2^2 \mathrm{d}t$$
$$\text{s.t. } \mathrm{d}\boldsymbol{x}_t^u = \left[ \theta_t(\boldsymbol{x}_1 - \boldsymbol{x}_t^u) + g_t \mathbf{u}_t \right] \mathrm{d}t, \boldsymbol{x}_0^u = \boldsymbol{x}_0, \boldsymbol{x}_1^u = \boldsymbol{x}_1. \quad (9)$$

UniDB underscores the equivalence between the SOC formulation under this limiting behavior and Doob's $h$-transform (Särkkä & Solin, 2019). By solving the problem (8), the closed-form optimal controller $\mathbf{u}_{t,\gamma}^*$ can be obtained. For more details, please refer to (Zhu et al., 2025).

## 4. Comparative Theoretical Framework for Diffusion Bridge and Flow Matching

Recent research (Gao et al., 2025) has mainly clarified the exact equivalence between diffusion models and flow matching (for the special case that the source distribution corresponds to a Gaussian). However, there remains confusion

about the connections between diffusion bridges and flow matching, since both two can achieve the transition between two arbitrary paired distributions. In the following analysis, we adopt the forward process of the UniDB-GOU model (denoted DB hereafter) and the well-known Flow Matching model (denoted FM hereafter) introduced in Section 3 above as the main diffusion bridge and flow matching model, respectively.

## 4.1. Connections between Diffusion Bridges and Flow Matching

To compare the two models within the same theoretical framework, we first consider the construction of the FM-related SOC problem to be consistent with the formulation of DB. Taking the substitution $\mathbf{u}_t = \dot{\eta}_t$ and $\boldsymbol{x}_t^u = \eta_t$ in the variational problem (4) leading to

$$\min_{\mathbf{u}_t} \int_0^1 \frac{1}{2}\|\mathbf{u}_t\|_2^2 \mathrm{d}t \tag{10}$$
$$\text{s.t. } \mathrm{d}\boldsymbol{x}_t^u = \mathbf{u}_t \mathrm{d}t, \boldsymbol{x}_0^u = \boldsymbol{x}_0, \boldsymbol{x}_1^u = \boldsymbol{x}_1,$$

with the optimal controller

$$\mathbf{u}_t^{*,\mathrm{FM}} = \frac{\mathrm{d}\eta_t^*}{\mathrm{d}t} = \boldsymbol{x}_1 - \boldsymbol{x}_0 = \frac{\boldsymbol{x}_1 - \boldsymbol{x}_t^u}{1-t}, \tag{11}$$

where the last equation comes from the interpolation (5) (Lipman et al., 2024). Despite the derivation of this FM's SOC problem, a direct comparison between the two SOC frameworks requires addressing a fundamental discrepancy in the formulation of their optimization objectives: the objective of the SDE-based SOC problem involves the expectation over all stochastic trajectories, whereas the ODE-based one does not, due to its determinism (Kappen, 2008). Considering that 1) the derivation of DB's optimal controller often relies on the ODE-based SOC formulation by the certainty equivalence principle (Chen et al., 2023; Rout et al., 2024; Zhu et al., 2025) in practice and 2) FM (10) forces exact target matching with $\boldsymbol{x}_1^u = \boldsymbol{x}_1$ which is the same as in (9), we adopt the ODE-based SOC problem (9) of DB to facilitate a fair comparison with the same ODE-based formulation of FM (10).

We can easily find the relation between (9) and (10) that the FM's SOC problem is a special case of DB's one, which leads to the following proposition:

**Proposition 4.1.** *Under the conditions that $\theta_t \to 0$ and $g_t = 1$ in (9), Diffusion Bridge degrades to Flow Matching.*

Details are provided in the Appendix A.1. This proposition indicates that under specific parameter constraints—namely, zero drift coefficient $\theta_t$ and unit diffusion coefficient $g_t$—the SOC formulation of DB (9) reduces to that of FM (10). The key distinction lies in the structure of the drift term: DB incorporates the drift term of the form $\theta_t(\boldsymbol{x}_1 - \boldsymbol{x}_t^u)$, which is

absent in FM. We further analyze the role of this drift term and present the following theorem to demonstrate that this drift term contributes to reducing the total cost (objective function) in the SOC problem.

**Theorem 4.2.** *Denote $\mathcal{J}(\mathbf{u}_t) \triangleq \int_0^1 \frac{1}{2}\|\mathbf{u}_t\|_2^2 \mathrm{d}t$ as the overall cost of the SOC problems (9) and (10) with the related optimal controller $\mathbf{u}_t^{*,\mathrm{DB}}$ and $\mathbf{u}_t^{*,\mathrm{FM}}$, respectively. Under the condition of diffusion coefficient $g_t = 1$ in (9) to be consistent with FM (10), then*

$$\mathcal{J}\left(\mathbf{u}_t^{*,\mathrm{DB}}\right) \leq \mathcal{J}\left(\mathbf{u}_t^{*,\mathrm{FM}}\right). \tag{12}$$

Please refer to Appendix A.2 for detailed proof. Theorem 4.2 mainly emphasizes that the drift term $\theta_t(\boldsymbol{x}_1 - \boldsymbol{x}_t^u)$ introduced in DB actively guides the system toward more stable trajectories, thereby lowering the total cost in the SOC problem. Lower total costs typically lead to smoother and more natural SDE/ODE trajectories. On one hand, a larger total cost implies the larger controller $\|\mathbf{u}_t^*\|_2^2$, which in turn excites oscillations along the forward trajectory and may disrupt the inherent continuity and smoothness of images. Consequently, the state $\boldsymbol{x}_t$ undergoes violent fluctuations at every time step, destabilizing the evolution of each individual pixel and inevitably degrading the visual quality of the generated image. On the other hand, from the SOC viewpoint, FM employs the trivial forward ODE, which may be too simple to characterise the transition between arbitrary distributions. As the discrepancy between the terminal distributions grows (e.g., in Section 5.3 (Experiments under Varying Levels of Task Difficulty), the masked center region in the Inpainting task gradually increases from 50×50 to 128×128, more blank spaces in the image center), the neural network becomes harder to fit the trajectories. However, DB incorporates the drift term $\theta_t(\boldsymbol{x}_1 - \boldsymbol{x}_t^u)$, which helps to construct a more stable transition between two arbitrary distributions.

## 4.2. Oversimplification and Ineffectiveness of FM's Interpolation Coefficient

In this section, we examine from an Optimal Transport (OT) perspective that the interpolation coefficients $t$ and $1-t$ in (5) of FM represent an oversimplification of McCann's approach, and it would become ineffective when the training data size gradually decreases.

**Oversimplification.** Under the conditions where the Brenier Theorem holds (Santambrogio, 2015; Lei & Gu, 2021) (the source measure is absolutely continuous), the McCann interpolation in OT (Villani, 2021) defines a deterministic dynamical system via the OT map $T : \Omega \to \Omega$, such that $T_{\#}x_0 = x_1$. Its Lagrangian trajectory and the corresponding Eulerian velocity field are given by:

$$v_{\mathrm{OT}}(x, t) = T(\tilde{x}_0(x, t)) - \tilde{x}_0(x, t), \tag{13}$$

*Table 1.* Quantitative results for Flow Matching and Diffusion Bridge (denoted FM and DB in table, respectively) under different image restoration and translation tasks.

| Method | Inpainting | | | | Deblurring | | | | 4×Super-Resolution | | | |
|---|---|---|---|---|---|---|---|---|---|---|---|---|
| | PSNR↑ | SSIM↑ | LPIPS↓ | FID↓ | PSNR↑ | SSIM↑ | LPIPS↓ | FID↓ | PSNR↑ | SSIM↑ | LPIPS↓ | FID↓ |
| FM | 23.54 | **0.760** | 0.106 | 17.84 | **24.67** | **0.683** | 0.228 | 38.18 | 27.11 | **0.789** | 0.088 | 11.61 |
| DB | **23.57** | 0.741 | **0.078** | **7.71** | 24.13 | 0.661 | **0.172** | **19.03** | **27.47** | 0.762 | **0.077** | **8.50** |

| Method | Edges→Handbags | | | | Edges→Shoes | | | | Nights→Days | | | |
|---|---|---|---|---|---|---|---|---|---|---|---|---|
| | MSE↓ | IS↑ | LPIPS↓ | FID↓ | MSE↓ | IS↑ | LPIPS↓ | FID↓ | MSE↓ | IS↑ | LPIPS↓ | FID↓ |
| FM | 0.130 | 2.94 | **0.189** | 29.24 | **0.034** | **3.57** | **0.176** | 71.38 | **0.114** | 3.52 | 0.423 | 121.53 |
| DB | **0.102** | **2.97** | 0.232 | **14.84** | 0.100 | 2.97 | 0.228 | **20.51** | 0.139 | **4.18** | **0.377** | **38.28** |

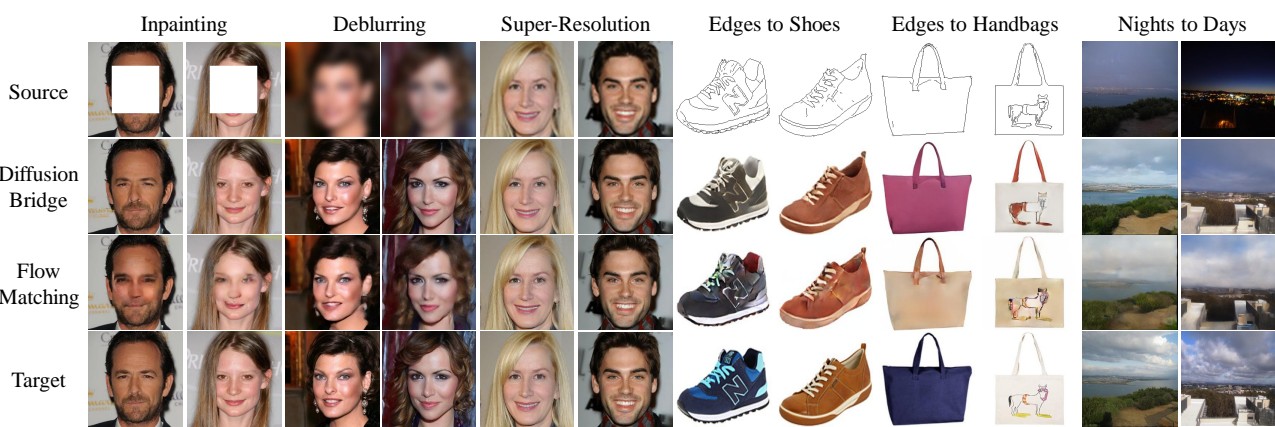

*Figure 2.* Qualitative comparison of visual results between Diffusion Bridge and Flow Matching on different Image Restoration tasks including Inpainting (centered 128×128 box mask), Deblurring (61×61 Gaussian kernel), 4×Super-Resolution, Edges→Handbags, Edges→Shoes, and Nights→Days.

where $\tilde{x}_0(x, t)$ is implicitly defined by the equation $x = (1-t)x_0 + tT(x_0)$, representing a complex inverse problem. In contrast, Flow Matching (FM) posits oversimplification by replacing the globally optimal coupling $T$ with a probabilistic coupling $q(x_0, x_1)$. For a sample $(x_0, x_1) \sim q$, FM defines a conditional flow and its velocity field as:

$$v_{\text{CFM}}(x(t) \mid x_0, x_1) = x_1 - x_0. \quad (14)$$

The genuine McCann interpolant of OT allows the velocity field to vary with $t$ while still satisfying the continuity equation, thereby minimizing the total kinetic energy. In contrast, FM's fixed interpolation coefficients, $t$ and $1 - t$, freeze the velocity field, which remains viable under small distributional discrepancies but leads to rapid performance degradation as the discrepancy grows. This failure stems from its inability to capture the complex inter-manifold geometry, triggering a large number of streamline intersections and vector field conflicts, ultimately leading to blurry learning objectives and a significant decrease in generation quality of the model (Liu et al., 2022).

**Ineffective with scarce data.** In practical learning sce-

narios, the empirical measures are constructed from paired training data, which are discrete measures consisting of finitely many $n \in \mathbb{Z}^+$ sample points $\{(x_i, y_i)\}$ as

$$\hat{\mu}_0^{(n)} = \frac{1}{n} \sum_{i=1}^{n} \delta_{x_i}, \quad \hat{\mu}_1^{(n)} = \frac{1}{n} \sum_{i=1}^{n} \delta_{y_i}, \quad (15)$$

where $\hat{\mu}_0$ and $\hat{\mu}_1$ represent the source and target measures, respectively. This discrete setting diverges fundamentally from the continuous assumptions (the absolutely continuous source measure) discussed above in Brenier potential theorem, which leads to the following remark:

*Remark* 4.3. For the finite empirical measures, they violate the absolute-continuity assumption required by Brenier's theorem; consequently, the existence of a convex potential function pushing $\hat{\mu}_0^{(n)}$ to $\hat{\mu}_1^{(n)}$ cannot be guaranteed, and McCann's interpolation is no longer well-defined.

Please refer to Appendix A.3 for detailed derivation. Remark 4.3 shows that

- When training data are limited, the interpolation coeffi-

*Table 2.* Quantitative results for Flow Matching and Diffusion Bridge (denoted FM and DB in table, respectively) under Image Inpainting tasks with different center box masks ranging from 50×50 to 128×128 on the CelebA-HQ dataset.

| Method | Box50 | | Box64 | | Box72 | | Box80 | | Box96 | | Box128 | |
|---|---|---|---|---|---|---|---|---|---|---|---|---|
| | LPIPS↓ | FID↓ | LPIPS↓ | FID↓ | LPIPS↓ | FID↓ | LPIPS↓ | FID↓ | LPIPS↓ | FID↓ | LPIPS↓ | FID↓ |
| FM | **0.035** | **4.93** | 0.039 | 5.13 | 0.042 | 5.43 | 0.047 | 5.86 | 0.060 | 8.18 | 0.106 | 17.84 |
| DB | **0.035** | **4.93** | **0.038** | **5.11** | **0.041** | **5.25** | **0.044** | **5.34** | **0.052** | **6.25** | **0.078** | **7.71** |

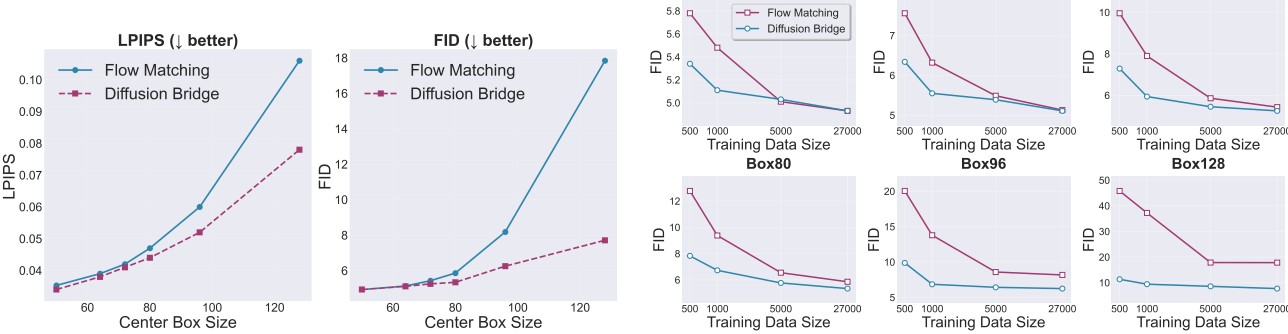

*(a)* Performance with increasing Inpainting mask sizes.  *(b)* Performance with varying training data sizes.

*Figure 3.* Comparison of Flow Matching and Diffusion Bridge on Image Inpainting tasks. (a) shows performance under different mask sizes; (b) shows performance under different training data sizes.

cients $t$ and $1-t$ lose their validity and the resulting path no longer corresponds to an OT interpolation.

- As the training data size grows ($n \to \infty$), then $\hat{\mu}_0^{(n)} \rightharpoonup \mu_0, \hat{\mu}_1^{(n)} \rightharpoonup \mu_1$ weakly and the empirical interpolation regains absolute continuity; McCann's interpolation is asymptotically restored, and the performance of FM improves. However, the coefficients remain rigidly fixed at $t$ and $1-t$, which cannot truly express the local velocity adjustment of optimal transmission. As the discrepancy between distributions widens and the task complexity escalates, the error rate climbs rapidly.

Simply adding more data reduces but cannot eliminate the systematic error of the frozen vector field. The drift term $\theta_t(\boldsymbol{x}_1 - \boldsymbol{x}_t^u)$, as incorporated by DB, is required to approximate the true Wasserstein geodesic in regimes of scarce data or complex deformation.

## 5. Comparative Experimental Evaluation of Diffusion Bridge versus Flow Matching

In this section, we evaluate the performance of Diffusion Bridge (DB) versus Flow Matching (FM) through various image-to-image tasks including Image Restoration tasks (Inpainting, Deblurring, and 4×Super-Resolution) on CelebA-HQ 256×256 (Karras, 2017) dataset and Image Translation tasks (Edges→Handbags (Isola et al., 2017), Edges→Shoes (Isola et al., 2017), and Nights→Days (Laffont et al., 2014))

in 256×256 resolutions.

For image restoration tasks, we take four common image evaluation metrics: Peak Signal-to-Noise Ratio (PSNR, higher is better) (Fardo et al., 2016), Structural Similarity Index (SSIM, higher is better) (Wang et al., 2004), Learned Perceptual Image Patch Similarity (LPIPS, lower is better) (Zhang et al., 2018), and Fréchet Inception Distance (FID, lower is better) (Heusel et al., 2017). For image translation tasks, we use Mean Squared Root (MSE, lower is better, in $[-1, 1]$ scale), Inception Scores (IS, higher is better) (Barratt & Sharma, 2018), LPIPS, and FID for evaluation metrics. For related implementation details, additional experimental and additional visual results, please refer to Appendix C, D, and E, respectively.

### 5.1. Experiment Setup: Same Transformer Architecture

FM has now widely adopted both U-Net (Lipman et al., 2024) and Transformer (Dao et al., 2023; Hu et al., 2024a) as its training network architecture, while the previous DB relied only on the U-Net network (Zhou et al., 2023; Yue et al., 2024; Zhu et al., 2025). In practice, substantial variations in U-Net implementations—particularly in parameter count and architectural details—make it difficult to conduct a direct comparative evaluation. To ensure a fair empirical comparison between DB and FM, we present a Latent Transformer-based Network for DB built on the DiT architecture (Peebles & Xie, 2023; Ma et al., 2024), unifying the network backbone of the two models. Please refer to

*Table 3.* Quantitative results for Flow Matching and Diffusion Bridge (denoted FM and DB in table, respectively) under the same training budget of Image Inpainting tasks (Box 128) on the CelebA-HQ dataset. The evaluation metric in the table is FID (lower is better).

| Method | 40 Epochs | 60 Epochs | 80 Epochs | 100 Epochs | 150 Epochs | 180 Epochs |
|--------|-----------|-----------|-----------|------------|------------|------------|
| FM | **19.31** | **18.32** | **18.57** | 18.26 | 18.64 | 18.57 |
| DB | 25.46 | 25.16 | 24.44 | **15.75** | **9.32** | **7.71** |

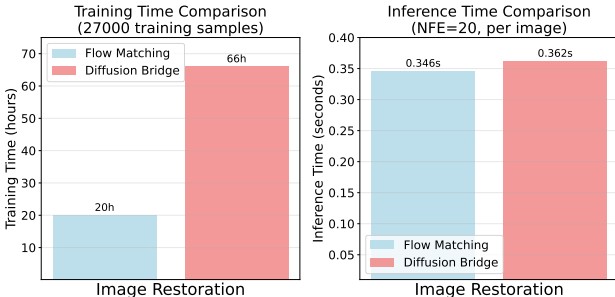

*Figure 4.* Training (with training data size 27000) and inference time (NFEs = 20) comparison on diffrent Image Restoration tasks (Inpainting, Deblurring, and Super-Resolution) between Diffusion Bridge and Flow Matching.

Appendix C for detailed parameters of the network.

Taking an image restoration task as an example, the overall training procedure can be summarized as follows. For paired Low-Quality (LQ) and High-Quality (HQ) images, denoted $\boldsymbol{x}^{\mathrm{LQ}}$ and $\boldsymbol{x}^{\mathrm{HQ}}$ with dimension $(h \times w \times 3)$, respectively, we map them into the latent space with dimension $(\frac{h}{8} \times \frac{w}{8} \times 4)$ using a pre-trained VAE encoder with a 8 down-sampling factor adopted from Stable Diffusion (Rombach et al., 2022). The latent codes are then input into the DiT block, which predicts either the score/noise (for DB) or the velocity (for FM) conditioned on the current time step, and the models are trained by minimizing the respective objective functions.

### 5.2. DB vs FM on Different Tasks

Here we demonstrate some experiments between the two models: Inpainting with centered 128×128 box mask, 4×Super-Resolution with bicubic down-sampling, Deblurring with 61×61 Gaussian kernel, Edges→Handbags, Edges→Shoes, and Nights→Days. Quantitative and qualitative results are shown in Table 1 and Figure 2. It can be concluded that DB consistently demonstrates significantly superior perceptual scores (LPIPS and FID) across all tasks, whereas FM exhibits relatively stronger performance on pixel-level metrics (SSIM and MSE). ODE-based models (FM) introduce zero additional noise during the sampling process, thereby generating much smoother images with better pixel-level distortion metrics (PSNR and SSIM). Meanwhile, according to Section 4.1, DB introduced the drift term in the SDE formulation, so its trajectories are more stable and better conform to the natural data manifold. There-

*Table 4.* Quantitative results among Flow Matching with two different network inputs $\boldsymbol{v}_\theta(\boldsymbol{z}_t, t)$ and $\boldsymbol{v}_\theta(\boldsymbol{z}_t, \boldsymbol{z}_T, t)$ and Diffusion Bridge (Denoted as FM w/ $\boldsymbol{v}_\theta(\boldsymbol{z}_t, t)$, FM w/ $\boldsymbol{v}_\theta(\boldsymbol{z}_t, \boldsymbol{z}_T, t)$, and DB w/ $\boldsymbol{\epsilon}_\theta(\boldsymbol{z}_t, \boldsymbol{z}_T, t)$, respectively) on three Inpainting task.

| Method | Box64 | | Box96 | | Box128 | |
|--------|-------|------|-------|------|--------|------|
| | LPIPS↓ | FID↓ | LPIPS↓ | FID↓ | LPIPS↓ | FID↓ |
| FM w/ $\boldsymbol{v}_\theta(\boldsymbol{z}_t, t)$ | 0.039 | 5.13 | 0.060 | 8.18 | 0.106 | 17.84 |
| FM w/ $\boldsymbol{v}_\theta(\boldsymbol{z}_t, \boldsymbol{z}_T, t)$ | 0.039 | 5.20 | 0.058 | 7.11 | 0.093 | 14.66 |
| DB w/ $\boldsymbol{\epsilon}_\theta(\boldsymbol{z}_t, \boldsymbol{z}_T, t)$ | **0.038** | **5.11** | **0.052** | **6.25** | **0.078** | **7.71** |

fore, the images generated by DB are visually more realistic and some unnatural patterns in smooth regions are avoided (better LPIPS and FID).

### 5.3. DB vs FM under Varying Levels of Task Difficulty

To assess the robustness of the two models under varying levels of task difficulty, we perform a series of Image Inpainting tasks with center box masks ranging from 50×50 to 128×128. Specifically, in image inpainting tasks, enlarging the center box mask amplifies the distance between the reference and target distributions, which represents the increasing levels of difficulty. We report the results of the LPIPS and FID scores of two models in Table 2. When the task is relatively straightforward (Box50 and Box64), FM achieves a performance comparable to that of DB. As the distributional discrepancy increases, FM exhibits markedly degraded stability compared to DB. Under this progressive shift, the two perceptual scores (LPIPS and FID) for FM, particularly in FID, increase significantly faster than those for DB, indicating a pronounced sensitivity to growing distributional shifts for FM and robustness on distribution discrepancy from the introduced drift term in DB, which validates the conclusion in Section 4.1.

### 5.4. DB vs FM with the Same Network Input

From the perspective of SDE/ODE modeling methodology, DB's network $\boldsymbol{\epsilon}_\theta(\boldsymbol{z}_t, \boldsymbol{z}_T, t)$ receives both the latent noisy interpolated state $\boldsymbol{z}_t$ and the latent low-quality condition $\boldsymbol{z}_T$ as input (Yue et al., 2024), whereas FM's network $\boldsymbol{v}_\theta(\boldsymbol{z}_t, t)$ only receives the noisy state $\boldsymbol{z}_t$ as input (Liu et al., 2022). Therefore, we modified the standard FM model by introducing $\boldsymbol{z}_T$ as an additional input into the velocity network $\boldsymbol{v}_\theta(\boldsymbol{z}_t, \boldsymbol{z}_T, t)$, thereby ensuring a more fair comparison. The results in Table 4 illustrate that although FM employs the

same neural network input conditions as DB, its performance, while improved, remains inferior to that of the DB. Therefore, the performance gap does not solely attribute to differences in network input conditions, but rather stems more fundamentally from the inherent mathematical distinctions in the underlying modeling of the two methods.

### 5.5. DB vs FM under Different Training Data Size

Further, we systematically conducted several Image Inpainting tasks on both DB and FM models under four different increasing training data sizes from 500, 1000, 5000, to 27000 to justify that the interpolation coefficients $t$ and $1-t$ in (5) of FM would become ineffective when the training data size gradually decreases, as mentioned in Section 4.2. The results are illustrated in Figure 3, which reveals that although the performance of both DB and FM deteriorates as training data become scarce, the degradation of FM is markedly steeper, while DB remains stable and maintains a consistently high performance level.

### 5.6. DB vs FM on Training and Inference Time

In addition, we conducted a systematic profiling of training and wall-clock inference latency for the tasks above. As depicted in Figure 4, under identical network architectures and experimental hyperparameter settings, after convergence, the training time of FM is substantially shorter than that of DB. We provide the quantitative results between DB and FM under the same training budget in Table 3. In the early stage, DB converges more slowly and delivers inferior results than FM. However, once FM has fully converged, DB continues to improve and ultimately achieves better performance. FM admits a simple, short, and direct gradient path that bypasses intermediate numerical complications, yielding superior numerical stability. In contrast, DB must predict a time-varying field, which is intrinsically difficult to fit, and gradients must back-propagate through a chain of compositional functions, leading to slower empirical convergence rate. Under the same generation conditions with the same Number of Function Evaluations (NFEs), the inference time of DB is basically similar to FM.

## 6. Conclusion

In this paper, we conduct a comprehensive analysis of Diffusion Bridge (DB) and Flow Matching (FM), the two most advanced methodologies currently available for distribution-to-distribution transformation. Under the unifying perspective of Stochastic Optimal Control (SOC), we establish that FM constitutes a special case of the DB framework, and we rigorously prove that the cost functional of FM is strictly larger than that minimized by DB because of the absence of the drift term in the forward process, thereby suggesting a less stable controlled trajectory. Concurrently, from the

perspective of Optimal Transport, we demonstrate that the linear interpolation coefficients $t$ and $1-t$ employed by FM violate the Brenier-potential theorem when the training data size is reduced, causing the algorithm to collapse and its performance to degrade precipitously. For the first time, we implement a Latent Transformer-based Diffusion Bridge and conduct experiments under the same architecture as Flow Matching, evaluating their performance through different image restoration and translation tasks, further confirming the respective advantages and disadvantages of the two models. Our theoretical research also has important reference significance for other fields, such as embodied AI, etc., on how to choose models that can achieve both performance and efficiency when training data size is small. Our future work may focus on comparing the performance of DB and FM in more general cases including high-resolution and generic text-to-image generation.

## Acknowledgment

This work was supported by the National Natural Science Foundation of China (62303319, 62406195), HPC Platform of ShanghaiTech University, and MoE Key Laboratory of Intelligent Perception and Human-Machine Collaboration (ShanghaiTech University), Shanghai Engineering Research Center of Intelligent Vision and Imaging. This work was also supported in part by computational resources provided by Fcloud CO., LTD.

## Impact Statement

There are many potential societal consequences of our work, none of which we feel must be specifically highlighted here.

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

# A. Proof

## A.1. Proof of Proposition 4.1

**Proposition 4.1.** *Under the conditions that $\theta_t \to 0$ and $g_t = 1$ in (9), Diffusion Bridge degrades to Flow Matching.*

*Proof.* Recall the DB's SOC problem (9):

$$\min_{\mathbf{u}_t} \int_0^1 \frac{1}{2}\|\mathbf{u}_t\|_2^2 \mathrm{d}t \tag{16}$$
$$\text{s.t. } \mathrm{d}\boldsymbol{x}_t^u = [\theta_t(\boldsymbol{x}_1 - \boldsymbol{x}_t^u) + g_t\mathbf{u}_t]\,\mathrm{d}t, \boldsymbol{x}_0^u = \boldsymbol{x}_0, \boldsymbol{x}_1^u = \boldsymbol{x}_1.$$

Taking conditions $\theta_t \to 0$ and $g_t = 1$, then DB's SOC problem (9) becomes

$$\min_{\mathbf{u}_t} \int_0^1 \frac{1}{2}\|\mathbf{u}_t\|_2^2 \mathrm{d}t \tag{17}$$
$$\text{s.t. } \mathrm{d}\boldsymbol{x}_t^u = \mathbf{u}_t\mathrm{d}t, \boldsymbol{x}_0^u = \boldsymbol{x}_0, \boldsymbol{x}_1^u = \boldsymbol{x}_1,$$

which is exactly equivalent to (10). $\square$

## A.2. Proof of Theorem 4.2

**Theorem 4.2.** *Denote $\mathcal{J}(\mathbf{u}_t) \triangleq \int_0^1 \frac{1}{2}\|\mathbf{u}_t\|_2^2\,\mathrm{d}t$ as the overall cost of the SOC problems (9) and (10) with the related optimal controller $\mathbf{u}_t^{*,DB}$ and $\mathbf{u}_t^{*,FM}$, respectively. Under the condition of diffusion coefficient $g_t = 1$ in (9) to be consistent with FM (10), then*

$$\mathcal{J}\left(\mathbf{u}_t^{*,DB}\right) \le \mathcal{J}\left(\mathbf{u}_t^{*,FM}\right). \tag{12}$$

*Proof.* Recall the SOC problem (9) with $g_t = 1$ as

$$\min_{\mathbf{u}_t} \int_0^1 \frac{1}{2}\|\mathbf{u}_t\|_2^2\mathrm{d}t \tag{18}$$
$$\text{s.t. } \mathrm{d}\boldsymbol{x}_t^u = [\theta_t(\boldsymbol{x}_1 - \boldsymbol{x}_t^u) + \mathbf{u}_t]\,\mathrm{d}t, \boldsymbol{x}_0^u = \boldsymbol{x}_0, \boldsymbol{x}_1^u = \boldsymbol{x}_1.$$

Denote $\bar{\theta}_{s:t} = \int_s^t \theta_z dz$, $\bar{\theta}_t = \int_0^t \theta_z dz$ for simplification when $s = 0$ and $\bar{\sigma}_{s:t}^2 = \lambda^2(1 - e^{-2\bar{\theta}_{s:t}})$. Then the related optimal controller $\mathbf{u}_t^{*,DB}$ and the transition $\boldsymbol{x}_t^u$ derived from UniDB (Zhu et al., 2025) with conditions $\gamma \to \infty$ and $g_t = 1$ are

$$\mathbf{u}_t^{*,DB} = \frac{e^{-2\bar{\theta}_{t:1}}(\boldsymbol{x}_1 - \boldsymbol{x}_t^u)}{\bar{\sigma}_{t:1}^2}$$
$$\boldsymbol{x}_t^u = \xi_t\boldsymbol{x}_0 + (1 - \xi_t)\boldsymbol{x}_1, \ \xi_t = e^{-\bar{\theta}_t}\frac{\bar{\sigma}_{t:1}^2}{\bar{\sigma}_1^2}. \tag{19}$$

Take $\boldsymbol{x}_t^u$ into $\mathbf{u}_t^{*,DB}$, we have

$$\mathbf{u}_t^{*,DB} = \frac{e^{-2\bar{\theta}_{t:1}}(\boldsymbol{x}_1 - \boldsymbol{x}_t^u)}{\bar{\sigma}_{t:1}^2} = \frac{e^{-2\bar{\theta}_{t:1}}\xi_t(\boldsymbol{x}_1 - \boldsymbol{x}_0)}{\bar{\sigma}_{t:1}^2} = \frac{e^{\bar{\theta}_t}e^{-2\bar{\theta}_1}}{\bar{\sigma}_1^2}(\boldsymbol{x}_1 - \boldsymbol{x}_0). \tag{20}$$

As for FM, we have

$$\mathbf{u}_t^{*,FM} = \boldsymbol{x}_1 - \boldsymbol{x}_0, \tag{21}$$

which is directly obtained from (11). Recall that $2\lambda^2\theta_t = g_t^2$ in UniDB and we have $g_t = 1$ which implies $\theta_t = \frac{1}{2\lambda^2}$ and we

have $e^x - 1 \geq x$ for $\forall\, x \geq 0$, then

$$
\begin{aligned}
\mathcal{J}\left(\mathbf{u}_t^{*,\mathrm{DB}}\right) &= \frac{1}{2}\int_0^1 \|\mathbf{u}_t^{*,\mathrm{DB}}\|_2^2 \mathrm{d}t \\
&= \frac{1}{2}\int_0^1 \|\frac{e^{\bar{\theta}_t}e^{-2\bar{\theta}_1}(\boldsymbol{x}_1 - \boldsymbol{x}_0)}{\bar{\sigma}_1^2}\|_2^2 \mathrm{d}t \\
&= \frac{1}{2}\int_0^1 \frac{e^{2\bar{\theta}_t}e^{-4\bar{\theta}_1}}{\lambda^4(1 - e^{-2\bar{\theta}_1})^2}\|\boldsymbol{x}_1 - \boldsymbol{x}_0\|_2^2 \mathrm{d}t \\
&= \frac{\|\boldsymbol{x}_1 - \boldsymbol{x}_0\|_2^2}{2\lambda^4(e^{2\bar{\theta}_1} - 1)^2}\int_0^1 e^{\frac{t}{\lambda^2}}\mathrm{d}t \\
&= \frac{\|\boldsymbol{x}_1 - \boldsymbol{x}_0\|_2^2}{2\lambda^2(e^{\frac{1}{\lambda^2}} - 1)^2}(e^{\frac{1}{\lambda^2}} - 1) \\
&= \frac{\|\boldsymbol{x}_1 - \boldsymbol{x}_0\|_2^2}{2\lambda^2(e^{\frac{1}{\lambda^2}} - 1)} \\
&\leq \frac{\|\boldsymbol{x}_1 - \boldsymbol{x}_0\|_2^2}{2\lambda^2\frac{1}{\lambda^2}} \\
&= \frac{1}{2}\|\boldsymbol{x}_1 - \boldsymbol{x}_0\|_2^2 \\
&= \frac{1}{2}\int_0^1 \|\mathbf{u}_t^{*,\mathrm{FM}}\|_2^2 \mathrm{d}t = \mathcal{J}\left(\mathbf{u}_t^{*,\mathrm{FM}}\right),
\end{aligned}
\tag{22}
$$

which concludes the proof.

$\square$

## A.3. Recovery of Absolute Continuity in the Empirical Wasserstein Geodesic Limit

For the empirical measures

$$
\hat{\mu}_0^{(n)} = \frac{1}{n}\sum_{i=1}^n \delta_{x_i} \rightharpoonup \mu_0 \quad (n \to \infty), \quad \hat{\mu}_1^{(n)} = \frac{1}{n}\sum_{j=1}^n \delta_{y_j} \rightharpoonup \mu_1 \quad (n \to \infty),
\tag{23}
$$

As $n$ is limited, for every $t \in (0,1)$, $\dim_{\mathcal{H}}\left(\mathrm{spt}\,\hat{\mu}_t^{(n)}\right) = 0 < d$

As $n \to \infty$ with $\hat{\mu}_0^{(n)} \rightharpoonup \mu_0, \hat{\mu}_1^{(n)} \rightharpoonup \mu_1$ weakly, one has

$$
\hat{\mu}_t^{(n)} \rightharpoonup \mu_t \quad \text{and} \quad \limsup_{n,m\to\infty} \dim_{\mathcal{H}}\left(\mathcal{S}_t^{(n)}\right) = d,
\tag{24}
$$

where $\mathcal{S}_t^{(n)} = \mathrm{spt}\,\hat{\mu}_t^{(n)} = \{(1-t)x_i + tT_n(x_i)\}_{i=1}^n$, $\mathrm{spt}\,\mu$ is the minimal closed support of the measure $\mu$, $\dim_{\mathcal{H}} E$ is the Hausdorff dimension of the set $E$.

*Proof.* Let

$$
\mu_0, \mu_1 \in \mathcal{P}_2^{\mathrm{ac}}\left(\mathbb{R}^d\right) \quad \text{with} \quad \mu_0, \mu_1 \ll \mathcal{L}^d,
\tag{25}
$$

and let

$$
\mu_t := [(1-t)\mathrm{id} + t\nabla\varphi]_\# \mu_0, \quad 0 \leq t \leq 1,
\tag{26}
$$

Suppose only finite samples are available:

$$
\hat{\mu}_0^{(n)} = \frac{1}{n}\sum_{i=1}^n \delta_{x_i} \rightharpoonup \mu_0 \quad (n \to \infty), \quad \hat{\mu}_1^{(n)} = \frac{1}{n}\sum_{j=1}^n \delta_{y_j} \rightharpoonup \mu_1 \quad (n \to \infty),
\tag{27}
$$

Assume, for contradiction, that $\hat{\mu}_0^{(n)} \ll \mathcal{L}^d$. Then there would exist $f \in L^1\left(\mathcal{L}^d\right)$ such that

$$\hat{\mu}_0^{(n)}(A) = \int_A f d\mathcal{L}^d \quad \forall A \in \mathscr{B}\left(\mathbb{R}^d\right). \tag{28}$$

Choosing $A = \{x_i\}$ yields $\mathcal{L}^d(A) = 0$ but $\hat{\mu}_0^{(n)}(A) = 1/n > 0$, contradicting absolute continuity.

Meanwhile, the support of any empirical measure is a finite set

$$\mathrm{spt}\left(\hat{\mu}_0^{(n)}\right) = \{x_1, \ldots, x_n\}, \quad \dim_{\mathcal{H}}\left(\mathrm{spt}\left(\hat{\mu}_0^{(n)}\right)\right) = 0 < d. \tag{29}$$

If $\mu_t \ll \mathcal{L}^d$, Caffarelli's interior regularity of optimal transport implies that $\nabla\varphi$ is a locally Hölder-homeomorphism; hence

$$\mathrm{spt}\,\mu_t = \overline{\{(1-t)x + t\nabla\varphi(x) : x \in \mathrm{spt}\,\mu_0\}}, \tag{30}$$

where the symbol $\bar{A}$ denotes the topological closure of the set $A$, that is, the union of $A$ with the set of all its limit points in the ambient space. Eq. (30) has full dimension:

$$\dim_{\mathcal{H}}\left(\mathrm{spt}\,\mu_t\right) = d. \tag{31}$$

As the empirical supports $S_t^{(n)}$ converge weakly to $\mathrm{spt}\,\mu_t$, Caffarelli's regularity ensures the Hausdorff dimension of the limit superior set recovers the full dimension $d$, which consistent with absolute continuity in the large-sample limit, the empirical supports

$$\mathcal{S}_t^{(n)} := \mathrm{spt}\,\hat{\mu}_t^{(n)} = \{(1-t)x_i + tT_n\left(x_i\right)\}_{i=1}^n, \tag{32}$$

become dense in $\mathrm{spt}\,\mu_t$ as $n \to \infty$. More precisely, for every $\epsilon > 0$ and every compact $K \subset \mathrm{spt}\,\mu_t$,

$$\mathcal{L}^d\left(K \backslash B_\epsilon\left(\mathcal{S}_t^{(n)}\right)\right) \to 0, \tag{33}$$

where $B_\epsilon\left(\mathcal{S}_t^{(n)}\right)$ is the $\epsilon$-neighbourhood of the set $\mathcal{S}_t^{(n)}$. Consequently, the upper Minkowski dimension (and hence the Hausdorff dimension) of the limit superior set satisfies

$$\limsup_{n,m\to\infty} \dim_{\mathcal{H}}\left(\mathcal{S}_t^{(n)}\right) = d. \tag{34}$$

Thus absolute continuity is recovered in the limit, and the empirical interpolation becomes an honest Wasserstein geodesic.

$\square$

## A.4. Optimality in Non-Compact Spaces

In Section 5.1, we points out that FM and DB utilize VAEs to compress images into a latent space. Although no hard constraint forces the latent codes produced by the VAE to lie in a compact set, we prove that, even in non-compact spaces, McCann interpolation remains universally valid for arbitrary distributional transport.

Let $\mu \in \mathcal{P}(\mathbb{R}^d)$ satisfy $\int_{\mathbb{R}^d} \|x\|^2 \, d\mu(x) < \infty$ and let $u : \mathbb{R}^d \to \mathbb{R} \cup \{+\infty\}$ be convex and $\mu$ a.e. differentiable. For the quadratic cost

$$c(x, y) := \tfrac{1}{2}\|x - y\|^2 \tag{35}$$

define $T = \nabla u$. The McCann interpolantion

$$x_t = (1-t)x_0 + tT, \quad t \in [0, 1], \tag{36}$$

constitutes an optimal transport map in the Wasserstein-2 sense, and this conclusion remains valid even when the support of $\mu$ is non-compact (Lei & Gu, 2021).

This demonstrates that, as long as the source distribution possesses finite second moments, the linear trajectory of FM (an oversimplification of McCann's approach) coincides with the optimal transport map, without requiring any additional regularity conditions irrespective of compactness.

The Monge Problem (MP) with quadratic cost is equivalent to:

$$\sup \left\{ M(T) := \int_X \langle x, T(x) \rangle d\mu, \quad T_{\#}\mu = \nu \right\} \tag{37}$$

This can be transformed into the dual problem:

$$\inf \left\{ \int_X u d\mu + \int_Y u^* d\nu : u(x) + u^*(y) \geqslant \langle x, y \rangle \right\}. \tag{38}$$

A convex function $u$ satisfies

$$u(x) + u^*(y) \geq \langle x, y \rangle \quad \forall x, y \in \mathbb{R}^d, \quad u(x) + u^*(y) = \langle x, y \rangle \quad \text{if} \quad y = \nabla u(x). \tag{39}$$

For any coupling $\gamma \in \Pi(\mu, \nu)$, we have

$$
\begin{aligned}
\int_{\mathbb{R}^d \times \mathbb{R}^d} \langle x, y \rangle d\gamma(x, y) &\leq \int_{\mathbb{R}^d \times \mathbb{R}^d} (u(x) + u^*(y)) \, d\gamma(x, y) \\
&= \int_{\mathbb{R}^d} u(x) d\mu(x) + \int_{\mathbb{R}^d} u^*(T(x)) d\mu(x) \\
&= \int_{\mathbb{R}^d} \langle x, T(x) \rangle d\mu(x).
\end{aligned}
\tag{40}
$$

Hence

$$\int_{\mathbb{R}^d \times \mathbb{R}^d} \langle x, y \rangle d\gamma \leq \int_{\mathbb{R}^d \times \mathbb{R}^d} \langle x, y \rangle d\gamma_T. \tag{41}$$

Moreover,

$$\int_{\mathbb{R}^d \times \mathbb{R}^d} \frac{1}{2} \left( \|x\|^2 + \|y\|^2 \right) d\gamma = \int_{\mathbb{R}^d \times \mathbb{R}^d} \frac{1}{2} \left( \|x\|^2 + \|y\|^2 \right) d\gamma_T. \tag{42}$$

Subtracting the two identities yields

$$\int_{\mathbb{R}^d \times \mathbb{R}^d} \frac{1}{2} \|x - y\|^2 d\gamma \geq \int_{\mathbb{R}^d \times \mathbb{R}^d} \frac{1}{2} \|x - y\|^2 d\gamma_T, \tag{43}$$

which establishes the optimality of $T$.

## B. Training and Inference Algorithms for DB and FM.

Here we demonstrate the training and inference algorithms we used in experiments for DB and FM. We take the training algorithm of UniDB with $\gamma \to \infty$ for DB (Yue et al., 2024) and the training objective (2) as the training algorithm for FM. As for the inference method of FM, we directly take the Euler sampling algorithm. We've learned that the Euler sampling method for FM is equivalent to DDPMs with DDIMs sampler (Gao et al., 2025). Hence, to ensure a fair comparison, we take the first-order UniDB++ (Pan et al., 2025) with $\gamma \to \infty$ (accelerating algorithms for GOUB, similar to DDIMs accelerating for DDPMs) as the inference method of DB.

---

**Algorithm 1** DB Training Algorithm

---

**repeat**

  Take a pair of images $\mathbf{x}_0$ and $\mathbf{x}_T$

  Encode the images as $\mathbf{z}_0 = \text{Encoder}(\mathbf{x}_0)$ and $\mathbf{z}_T = \text{Encoder}(\mathbf{x}_T)$

  $t \sim \text{Uniform}(\{1, ..., T\})$ and $\epsilon \sim \mathcal{N}(0, I)$

  $a = \frac{e^{-\bar{\theta}_{t-1:t}} \bar{\sigma}_{t:T}^2}{\bar{\sigma}_{t-1:T}^2}, b = \frac{1}{\bar{\sigma}_T^2}\left((1 - e^{-\bar{\theta}_t})\bar{\sigma}_{t:T}^2 + e^{-2\bar{\theta}_{t:T}}\bar{\sigma}_t^2 - ((1 - e^{-\bar{\theta}_{t-1}})\bar{\sigma}_{t-1:T}^2 + e^{-\bar{\theta}_{t-1:T}}\bar{\sigma}_{t-1}^2)a\right)$

  $\mathbf{z}_t = e^{-\bar{\theta}_t \frac{\bar{\sigma}_{t:T}^2}{\bar{\sigma}_T^2}}\mathbf{z}_0 + \left(1 - e^{-\bar{\theta}_t \frac{\bar{\sigma}_{t:T}^2}{\bar{\sigma}_T^2}}\right)\mathbf{z}_T + \bar{\sigma}_t'\epsilon, \bar{\sigma}_t'^2 = \bar{\sigma}_t^2 \bar{\sigma}_{t:T}^2 / \bar{\sigma}_T^2$

  $\bar{\boldsymbol{\mu}}_t = e^{-\bar{\theta}_t \frac{\bar{\sigma}_{t:T}^2}{\bar{\sigma}_T^2}}\mathbf{z}_0 + \left(1 - e^{-\bar{\theta}_t \frac{\bar{\sigma}_{t:T}^2}{\bar{\sigma}_T^2}}\right)\mathbf{z}_T$

  $\boldsymbol{\mu}_{t-1,\theta} = \mathbf{z}_t - \left(\theta_t + g_t^2 \frac{e^{-2\bar{\theta}_{t:T}}}{\bar{\sigma}_{t:T}^2}\right)(\mathbf{z}_T - \mathbf{z}_t) + \frac{g_t^2}{\bar{\sigma}_t'^2}\boldsymbol{\epsilon}_\theta(\mathbf{z}_t, \mathbf{z}_T, t)$

  $\boldsymbol{\mu}_{t-1} = \frac{1}{\bar{\sigma}_t'}[\bar{\sigma}_t'^2(\mathbf{x}_t - b\mathbf{x}_T)a + (\bar{\sigma}_t'^2 - \bar{\sigma}_{t-1}'^2 a^2)\bar{\boldsymbol{\mu}}_t]$

  Take gradient descent step on $\mathcal{L}_\theta = \mathbb{E}\left[\|\boldsymbol{\mu}_{t-1,\theta} - \boldsymbol{\mu}_{t-1}\|\right]$

**until** converged

---

**Algorithm 2** DB Inference Algorithm

---

**Input:** LQ images $\mathbf{x}_T$, noise predicted model $\boldsymbol{\epsilon}_\theta(\mathbf{x}_t, \mathbf{x}_T, t)$, and $M + 1$ time steps $\{t_i\}_{i=0}^{M}$ decreasing from $t_0 = T$ to $t_M = 0$. Initialize $\mathbf{z}_{t_0} = \text{Encoder}(\mathbf{x}_T)$.

**for** $i = 1$ **to** $M$ **do**

  Sample $\boldsymbol{\epsilon} \sim \mathcal{N}(0, I)$ if $i < M$, else $\boldsymbol{\epsilon} = 0$.

  $\kappa_0 = e^{\bar{\theta}_T}(1 - e^{-2\bar{\theta}_T}), \kappa_{t_{i-1}} = e^{\bar{\theta}_{t_{i-1}:T}}(1 - e^{-2\bar{\theta}_{t_{i-1}:T}}), \kappa_{t_i} = e^{\bar{\theta}_{t_i:T}}(1 - e^{-2\bar{\theta}_{t_i:T}})$

  $\rho_{t_{i-1}} = e^{\bar{\theta}_{t_{i-1}}}(1 - e^{-2\bar{\theta}_{t_{i-1}}}), \rho_{t_i} = e^{\bar{\theta}_{t_i}}(1 - e^{-2\bar{\theta}_{t_i}}), \delta_{t_{i-1}:t_i}^d = \lambda \rho_{t_i}\sqrt{\frac{1}{e^{2\bar{\theta}_{t_i}} - 1} - \frac{1}{e^{2\bar{\theta}_{t_{i-1}}} - 1}}$

  $\hat{\mathbf{z}}_{0|t_{i-1}} = (\mathbf{z}_{t_{i-1}} - (1 - e^{-\bar{\theta}_{t_{i-1}}}\frac{\bar{\sigma}_{t_{i-1}:T}^2}{\bar{\sigma}_T^2})\mathbf{z}_T - \bar{\sigma}_{t_{i-1}}'\boldsymbol{\epsilon}_\theta(\mathbf{x}_{t_{i-1}}, \mathbf{x}_T, t_{i-1}))/e^{-\bar{\theta}_{t_{i-1}}}\frac{\bar{\sigma}_{t_{i-1}:T}^2}{\bar{\sigma}_T^2}$.

  $\mathbf{z}_{t_i} = \frac{\rho_{t_i}}{\rho_{t_{i-1}}}\mathbf{z}_{t_{i-1}} + \left(1 - \frac{\rho_{t_i}}{\rho_{t_{i-1}}} + \frac{\kappa_{t_{i-1}}\rho_{t_i}}{\kappa_0 \rho_{t_{i-1}}} - \frac{\kappa_{t_i}}{\kappa_0}\right)\mathbf{z}_T + \left(\frac{\kappa_{t_i}}{\kappa_0} - \frac{\kappa_{t_{i-1}}\rho_{t_i}}{\kappa_0 \rho_{t_{i-1}}}\right)\hat{\mathbf{z}}_{0|t_{i-1}} + \delta_{t_{i-1}:t_i}^d \boldsymbol{\epsilon}$.

**end for**

$\tilde{\mathbf{x}}_0 = \text{Decoder}(\mathbf{z}_0)$

**Return** HQ images $\tilde{\mathbf{x}}_0$

---

**Algorithm 3** FM Training Algorithm

---

**repeat**

  Take a pair of images $\mathbf{x}_0$ and $\mathbf{x}_1$

  Encode the images as $\mathbf{z}_0 = \text{Encoder}(\mathbf{x}_0)$ and $\mathbf{z}_1 = \text{Encoder}(\mathbf{x}_1)$

  $t \sim \text{Uniform}(\{\frac{1}{T}, ..., \frac{T}{T} = 1\})$

  $\mathbf{z}_t = (1 - t)\mathbf{z}_0 + t\mathbf{z}_1$

  Take gradient descent step on $\mathcal{L}_\theta = \mathbb{E}\left[\|\boldsymbol{v}_\theta(\mathbf{z}_t, t) - (\mathbf{z}_1 - \mathbf{z}_0)\|\right]$

**until** converged

---

**Algorithm 4** FM Inference Algorithm

---

**Input:** LQ images $\mathbf{x}_1$, velocity predicted model $\boldsymbol{v}_\theta(\mathbf{x}_t, t)$, and $M + 1$ time steps $\{t_i\}_{i=0}^{M}$ decreasing from $t_0 = 1$ to $t_M = 0$. Initialize $\mathbf{z}_{t_0} = \text{Encoder}(\mathbf{x}_1)$.

**for** $i = 1$ **to** $M$ **do**

  $\mathbf{z}_{t_i} = \mathbf{z}_{t_{i-1}} + (t_{i-1} - t_i)\boldsymbol{v}_\theta(\mathbf{z}_{t_{i-1}}, t_{i-1})$

**end for**

$\tilde{\mathbf{x}}_0 = \text{Decoder}(\mathbf{z}_0)$

**Return** HQ images $\tilde{\mathbf{x}}_0$

---

# C. Experimental and Implementation Details

**Datasets**. We list details about the datesets used in all image restoration and translation tasks in Table 5.

*Table 5.* Details about the used datasets in all image restoration tasks.

| Task | Dataset | Number of Training Images | Number of Testing Images |
|---|---|---|---|
| Inpainting | CelebA-HQ | 500/1000/5000/27000 | 3000 |
| Inpainting | FFHQ | 69000 | 1000 |
| Super-Resolution, Deblurring | CelebA-HQ | 27000 | 3000 |
| Style Transfer | CelebA-HQ | 27000 | 3000 |
| Image Translation | CelebAMask-HQ | 26000 | 2000 |
| | Edges→Handbags | 10000 | 10000 |
| | Edges→Shoes | 10000 | 10000 |
| | Nights→Days | 10000 | 10000 |

**Unified Transformer Architecture for Diffusion Bridge and Flow Matching**. We list some shared Transformer Hyper-parameters in Table 6.

*Table 6.* Shared Transformer Hyper-parameters.

| Hyper-parameter | Value |
|---|---|
| Patch size | 2 |
| Hidden size | 1024 |
| Depth | 24 |
| Attention heads | 16 |
| MLP ratio | 4.0 |

**Transformer Architecture v.s. Standard DiT**. FM has extensively leveraged Latent Transformers for text-to-image generation and image restoration tasks, as exemplified by CrossFlow (Liu et al., 2025) and FM in Latent Space (Dao et al., 2023). In contrast, DB has yet to adopt such powerful neural architectures. Compared with the standard DiT, whose forward function takes $(\boldsymbol{x}_t, t)$ as input, the forward function of our DB Transformer takes $(\boldsymbol{x}_t, \boldsymbol{x}_T, t)$ as input, where $\boldsymbol{x}_t$ and $\boldsymbol{x}_T$ are combined via subtraction and concatenation operations. Specifically, we first compute $\boldsymbol{x}_t - \boldsymbol{x}_T$, then concatenate it with $\boldsymbol{x}_T$ along the channel dimension.

**All Hyper-parameters.** We set steady variance level $\lambda^2 = 30^2/255^2$, coefficient $e^{-\bar{\theta}_T} = 0.005$ instead of zero, 8 batch size when training, ADAM optimizer with $\beta_1 = 0.9$ and $\beta_2 = 0.99$ (Kingma, 2014), 600 thousand total training steps with $10^{-4}$ initial learning rate and decaying by half at 300 and 500 thousand iterations. We choose a flipped version of cosine noise schedule for $\theta_t$ (Luo et al., 2023b),

$$\theta_t = 1 - \frac{\cos^2\left(\frac{t/T+s}{1+s}\frac{\pi}{2}\right)}{\cos^2\left(\frac{s}{1+s}\frac{\pi}{2}\right)}, \tag{44}$$

where $s = 0.008$ is followed from (Yue et al., 2024; Zhu et al., 2025; Pan et al., 2025) to achieve a smooth noise schedule. As for time schedule, we directly take the naive uniform time schedule.

**Implementation Details.** All experiments are trained and tested on a single NVIDIA H20 GPU with 141GB memory.

# D. Additional Experimental Results

## D.1. Different Inpainting Masks

Here we illustrate detailed experimental results on different inpainting masks like thin, thick, and every-second-line (ev2li) masks adopted from RePaint (Lugmayr et al., 2022) under different training data sizes in Table 7 and Table 8.

*Table 7.* Quantitative results for Flow Matching and Diffusion Bridge (denoted FM and DB in table, respectively) under different Image Inpainting tasks with different training data sizes.

| | Training data size 500 | | | | | | | | | | | |
|---|---|---|---|---|---|---|---|---|---|---|---|---|
| **Method** | **Box50** | | | | **Box64** | | | | **Box72** | | | |
| | PSNR↑ | SSIM↑ | LPIPS↓ | FID↓ | PSNR↑ | SSIM↑ | LPIPS↓ | FID↓ | PSNR↑ | SSIM↑ | LPIPS↓ | FID↓ |
| FM | 27.33 | **0.833** | 0.042 | 5.78 | 26.16 | **0.819** | 0.050 | 7.56 | 25.79 | **0.812** | **0.055** | 9.96 |
| DB | **28.04** | 0.813 | **0.038** | **5.34** | **26.43** | 0.790 | **0.047** | **6.34** | **26.54** | 0.779 | 0.056 | **7.29** |
| **Method** | **Box80** | | | | **Box96** | | | | **Box128** | | | |
| | PSNR↑ | SSIM↑ | LPIPS↓ | FID↓ | PSNR↑ | SSIM↑ | LPIPS↓ | FID↓ | PSNR↑ | SSIM↑ | LPIPS↓ | FID↓ |
| FM | 25.26 | **0.803** | 0.062 | 12.77 | 23.93 | **0.779** | 0.078 | 20.05 | 21.43 | **0.719** | 0.133 | 45.81 |
| DB | **25.45** | 0.777 | **0.058** | **7.83** | **23.95** | 0.753 | **0.070** | **9.88** | **21.70** | 0.705 | **0.103** | **11.34** |
| **Method** | **Thin mask** | | | | **Thick mask** | | | | **ev2li** | | | |
| | PSNR↑ | SSIM↑ | LPIPS↓ | FID↓ | PSNR↑ | SSIM↑ | LPIPS↓ | FID↓ | PSNR↑ | SSIM↑ | LPIPS↓ | FID↓ |
| FM | 18.73 | 0.588 | 0.239 | 69.47 | 17.95 | 0.678 | 0.200 | 66.40 | 24.27 | 0.681 | 0.178 | 39.56 |
| DB | **23.41** | **0.690** | **0.120** | **14.83** | **21.10** | **0.700** | **0.131** | **12.32** | **25.42** | **0.702** | **0.128** | **16.03** |
| | **Training data size 1000** | | | | | | | | | | | |
| **Method** | **Box50** | | | | **Box64** | | | | **Box72** | | | |
| | PSNR↑ | SSIM↑ | LPIPS↓ | FID↓ | PSNR↑ | SSIM↑ | LPIPS↓ | FID↓ | PSNR↑ | SSIM↑ | LPIPS↓ | FID↓ |
| FM | 27.62 | **0.836** | 0.047 | 5.48 | 26.68 | **0.824** | 0.047 | 6.32 | 26.18 | **0.816** | 0.052 | 7.90 |
| DB | **27.80** | 0.810 | **0.039** | **5.11** | **27.18** | 0.802 | **0.044** | **5.55** | **26.68** | 0.797 | **0.048** | **5.94** |
| **Method** | **Box80** | | | | **Box96** | | | | **Box128** | | | |
| | PSNR↑ | SSIM↑ | LPIPS↓ | FID↓ | PSNR↑ | SSIM↑ | LPIPS↓ | FID↓ | PSNR↑ | SSIM↑ | LPIPS↓ | FID↓ |
| FM | 25.62 | **0.807** | 0.058 | 9.38 | 24.34 | **0.785** | 0.073 | 13.79 | 21.76 | **0.724** | 0.124 | 37.23 |
| DB | **26.03** | 0.786 | **0.053** | **6.73** | **24.97** | 0.772 | **0.061** | **6.86** | **22.23** | 0.717 | **0.096** | **9.43** |
| **Method** | **Thin mask** | | | | **Thick mask** | | | | **ev2li** | | | |
| | PSNR↑ | SSIM↑ | LPIPS↓ | FID↓ | PSNR↑ | SSIM↑ | LPIPS↓ | FID↓ | PSNR↑ | SSIM↑ | LPIPS↓ | FID↓ |
| FM | 20.00 | 0.625 | 0.203 | 52.81 | 19.22 | 0.689 | 0.179 | 48.29 | 24.56 | 0.689 | 0.162 | 34.70 |
| DB | **24.13** | **0.711** | **0.103** | **11.45** | **21.96** | **0.712** | **0.120** | **9.98** | **26.01** | **0.729** | **0.113** | **13.07** |
| | **Training data size 5000** | | | | | | | | | | | |
| **Method** | **Box50** | | | | **Box64** | | | | **Box72** | | | |
| | PSNR↑ | SSIM↑ | LPIPS↓ | FID↓ | PSNR↑ | SSIM↑ | LPIPS↓ | FID↓ | PSNR↑ | SSIM↑ | LPIPS↓ | FID↓ |
| FM | **28.32** | 0.843 | **0.037** | **5.01** | 27.54 | **0.834** | 0.042 | 5.49 | **27.07** | **0.828** | 0.046 | 5.86 |
| DB | 28.04 | **0.881** | **0.037** | 5.03 | **27.62** | 0.809 | **0.041** | **5.39** | 27.04 | 0.802 | **0.045** | **5.45** |
| **Method** | **Box80** | | | | **Box96** | | | | **Box128** | | | |
| | PSNR↑ | SSIM↑ | LPIPS↓ | FID↓ | PSNR↑ | SSIM↑ | LPIPS↓ | FID↓ | PSNR↑ | SSIM↑ | LPIPS↓ | FID↓ |
| FM | **26.50** | **0.819** | 0.051 | 6.55 | 25.24 | **0.799** | 0.064 | 8.59 | 22.65 | **0.744** | 0.107 | 17.87 |
| DB | 26.19 | 0.794 | **0.048** | **5.77** | **25.39** | 0.778 | **0.058** | **6.43** | **22.96** | 0.733 | **0.085** | **8.59** |
| **Method** | **Thin mask** | | | | **Thick mask** | | | | **ev2li** | | | |
| | PSNR↑ | SSIM↑ | LPIPS↓ | FID↓ | PSNR↑ | SSIM↑ | LPIPS↓ | FID↓ | PSNR↑ | SSIM↑ | LPIPS↓ | FID↓ |
| FM | 23.38 | 0.713 | 0.129 | 23.69 | 21.35 | **0.725** | 0.146 | 24.52 | 25.66 | 0.724 | 0.126 | 19.99 |
| DB | **24.66** | **0.724** | **0.095** | **9.68** | **22.16** | 0.719 | **0.110** | **8.64** | **26.23** | **0.731** | **0.102** | **11.28** |

*Table 8.* Quantitative results for Flow Matching and Diffusion Bridge (denoted FM and DB in table, respectively) under different Image Inpainting tasks on the CelebA-HQ dataset.

| Method | Box50 | | | | Box64 | | | | Box72 | | | |
|---|---|---|---|---|---|---|---|---|---|---|---|---|
| | **PSNR↑** | **SSIM↑** | **LPIPS↓** | **FID↓** | **PSNR↑** | **SSIM↑** | **LPIPS↓** | **FID↓** | **PSNR↑** | **SSIM↑** | **LPIPS↓** | **FID↓** |
| FM | **28.66** | **0.847** | **0.035** | **4.93** | **28.03** | **0.840** | 0.039 | 5.13 | **27.68** | **0.835** | 0.042 | 5.43 |
| DB | **28.65** | 0.820 | **0.035** | **4.93** | 27.90 | 0.813 | **0.038** | **5.11** | 27.45 | 0.807 | **0.041** | **5.25** |

| Method | Box80 | | | | Box96 | | | | Box128 | | | |
|---|---|---|---|---|---|---|---|---|---|---|---|---|
| | **PSNR↑** | **SSIM↑** | **LPIPS↓** | **FID↓** | **PSNR↑** | **SSIM↑** | **LPIPS↓** | **FID↓** | **PSNR↑** | **SSIM↑** | **LPIPS↓** | **FID↓** |
| FM | **27.17** | **0.828** | 0.047 | 5.86 | **26.02** | **0.810** | 0.060 | 8.18 | 23.54 | **0.760** | 0.106 | 17.84 |
| DB | 27.08 | 0.802 | **0.044** | **5.34** | 25.85 | 0.786 | **0.052** | **6.25** | **23.57** | 0.741 | **0.078** | **7.71** |

| Method | Thin mask | | | | Thick mask | | | | ev2li | | | |
|---|---|---|---|---|---|---|---|---|---|---|---|---|
| | **PSNR↑** | **SSIM↑** | **LPIPS↓** | **FID↓** | **PSNR↑** | **SSIM↑** | **LPIPS↓** | **FID↓** | **PSNR↑** | **SSIM↑** | **LPIPS↓** | **FID↓** |
| FM | 24.68 | **0.746** | 0.105 | 14.90 | 22.51 | **0.748** | 0.129 | 15.82 | 25.69 | 0.731 | 0.122 | 19.53 |
| DB | **25.28** | 0.739 | **0.083** | **8.35** | **22.86** | 0.736 | **0.096** | **7.34** | **26.99** | **0.749** | **0.084** | **9.59** |

## D.2. Different Datasets

Here we illustrate more results between DB and FM on different inpainting tasks under the more challenging FFHQ 256×256 dataset (Karras et al., 2019) in Table 9. It shows that FM exhibits markedly degraded stability compared to DB as the task difficulty increases, which is consistent with our main experiments.

*Table 9.* Quantitative results for Flow Matching and Diffusion Bridge (denoted FM and DB in table, respectively) under different Image Inpainting tasks on the FFHQ dataset.

| Method | Box64 | | Box96 | | Box128 | |
|---|---|---|---|---|---|---|
| | **LPIPS↓** | **FID↓** | **LPIPS↓** | **FID↓** | **LPIPS↓** | **FID↓** |
| FM | 0.047 | 11.16 | 0.069 | 14.89 | 0.118 | 25.37 |
| DB | **0.044** | **11.11** | **0.059** | **12.02** | **0.089** | **16.72** |

## D.3. Different Sampling Steps

Here we provide a comparison between DB and FM under comparable inference time budgets (different NFEs). It can be observed that, even when FM and DB are allotted the same compute budget, the performance of FM is constrained by its own model capacity, which is still falls markedly short of that achieved by DB.

*Table 10.* Quantitative results for Flow Matching and Diffusion Bridge (denoted FM and DB in table, respectively) under different NFEs of Image Inpainting tasks (Box 128) on the CelebA-HQ dataset.

| Method | 10 NFEs | | 20 NFEs | | 50 NFEs | | 100 NFEs | |
|---|---|---|---|---|---|---|---|---|
| | **LPIPS↓** | **FID↓** | **LPIPS↓** | **FID↓** | **LPIPS↓** | **FID↓** | **LPIPS↓** | **FID↓** |
| FM | 0.111 | 23.01 | 0.106 | 17.84 | 0.099 | 17.40 | 0.101 | 17.51 |
| DB | **0.078** | **9.32** | **0.078** | **7.71** | **0.079** | **7.20** | **0.080** | **7.14** |

## D.4. Comparison with Some Baselines

Here we involve the results of GOUB and UniDB (with U-Net) on 4×Super-Resolution task on CelebA-HQ dataset as a baseline in Table 11. For super-resolution tasks, we mainly focus on the perceptual scores (LPIPS and FID) (Luo et al., 2023b). Our DB and FM (based on the 485M-parameter Transformer) significantly outperforms existing UniDB and GOUB (based on the 150M-parameter U-Net).

*Table 11.* Quantitative results between our Diffusion Bridge, our Flow Matching and U-Net-based diffusion bridge baselines (GOUB and UniDB).

| Method | Network Type | 4×Super-Resolution | | | |
|---|---|---|---|---|---|
| | | PSNR↓ | SSIM↓ | LPIPS↓ | FID↓ |
| GOUB | U-Net | 28.63 | 0.777 | 0.104 | 19.02 |
| UniDB | U-Net | **28.70** | **0.789** | 0.090 | 17.59 |
| FM | Latent Transformer (Ours) | 27.11 | **0.789** | 0.088 | 11.61 |
| DB | Latent Transformer (Ours) | 27.47 | 0.762 | **0.077** | **8.50** |

## D.5. Experimental results for $g_t = 1$ in DB

In Theorem 4.2, we made the assumptions $g_t = 1$ in DB to keep the formula the same as FM. Here we provide a comparison between DB with $g_t = 1$ and FM in Table 12 and the results align with Theorem 4.2. DB with $g_t = 1$ still shows better performance than FM. It confirms the observed stability is an intrinsic theoretical property of DB.

*Table 12.* Quantitative results for Flow Matching and Diffusion Bridge (denoted FM and DB in table, respectively) under different settings of Image Inpainting tasks on the CelebA-HQ dataset.

| Method | Box96 | | Box128 | |
|---|---|---|---|---|
| | LPIPS↓ | FID↓ | LPIPS↓ | FID↓ |
| FM | 0.060 | 8.18 | 0.106 | 17.84 |
| DB | **0.052** | **6.25** | **0.078** | **7.71** |
| DB ($g_t = 1$) | **0.052** | 5.96 | 0.079 | 7.86 |

# E. Additional Visual Experimental Results

Here we illustrate more visual results.

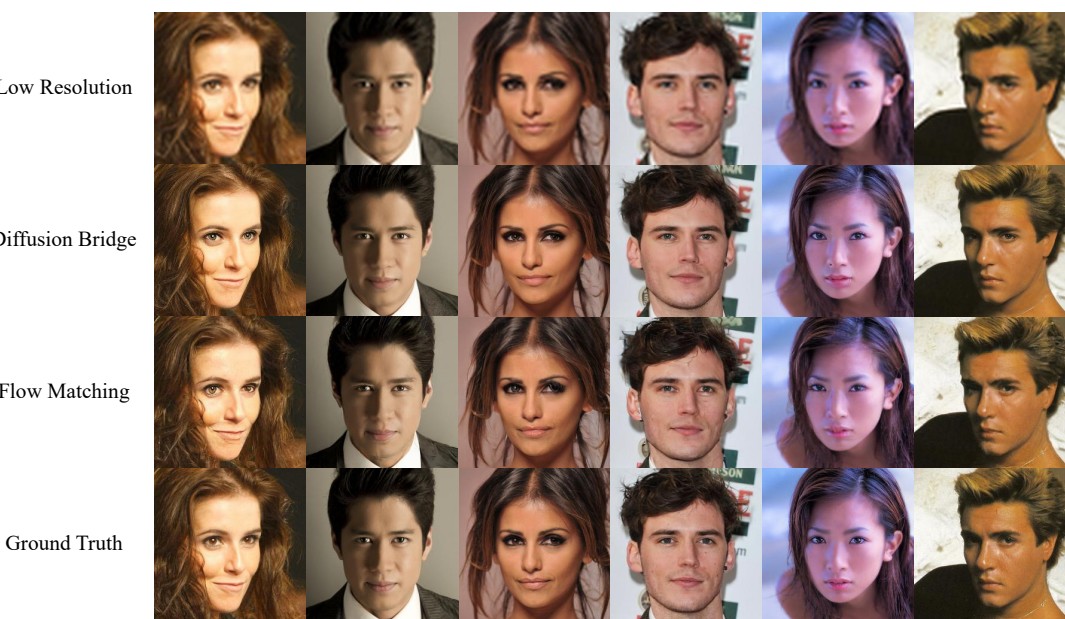

*Figure 5.* Additional visual comparison between Diffusion Bridge and Flow Matching on 4×Super-Resolution on the CelebFaces dataset.

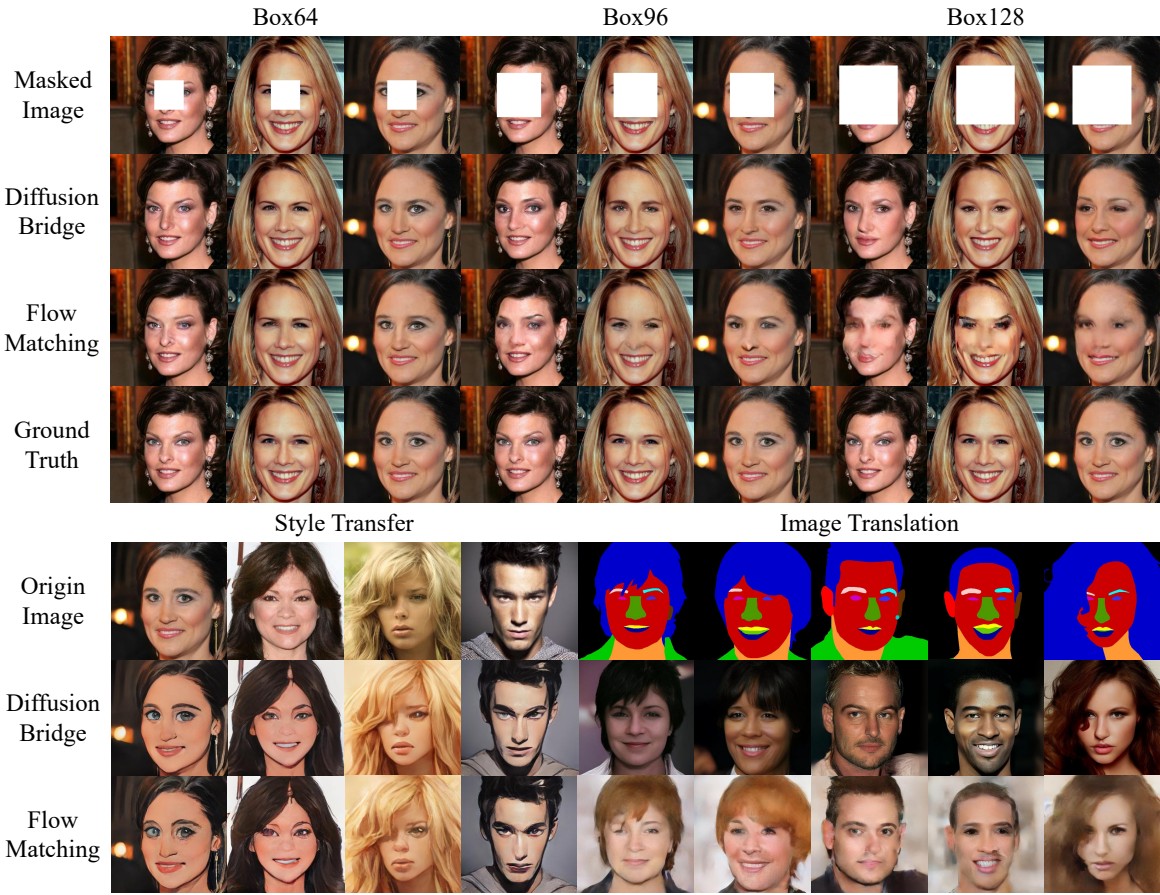

*Figure 6.* Qualitative comparison of visual results between Diffusion Bridge and Flow Matching in Image Inpainting, Style Transfer and Image Translation tasks.

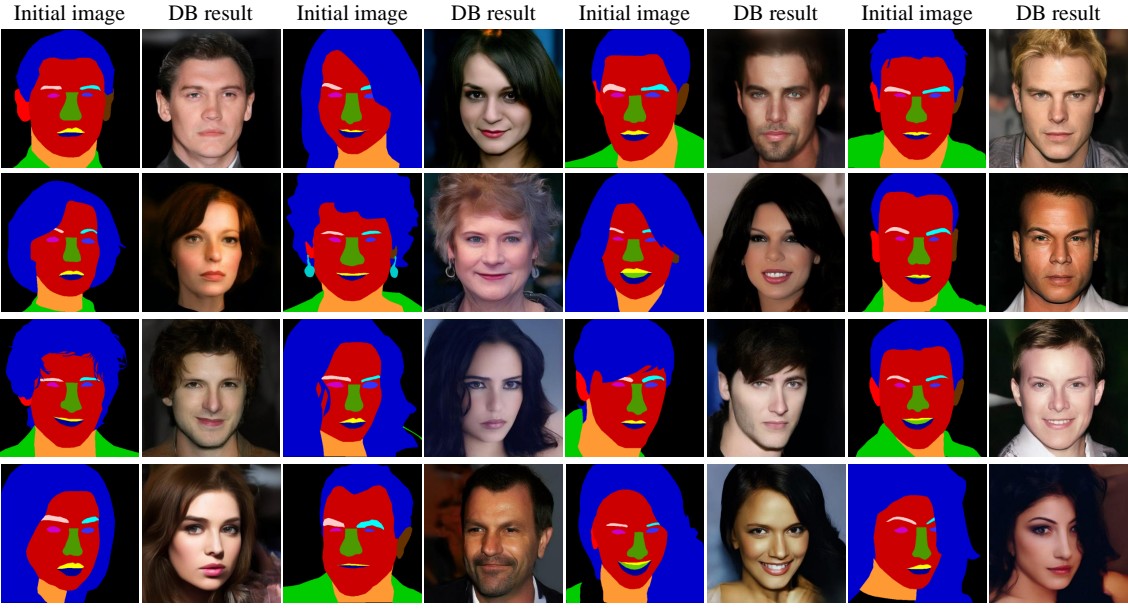

*Figure 7.* Additional Diffusion Bridge visual results on Image Translation with CelebFaces dataset.

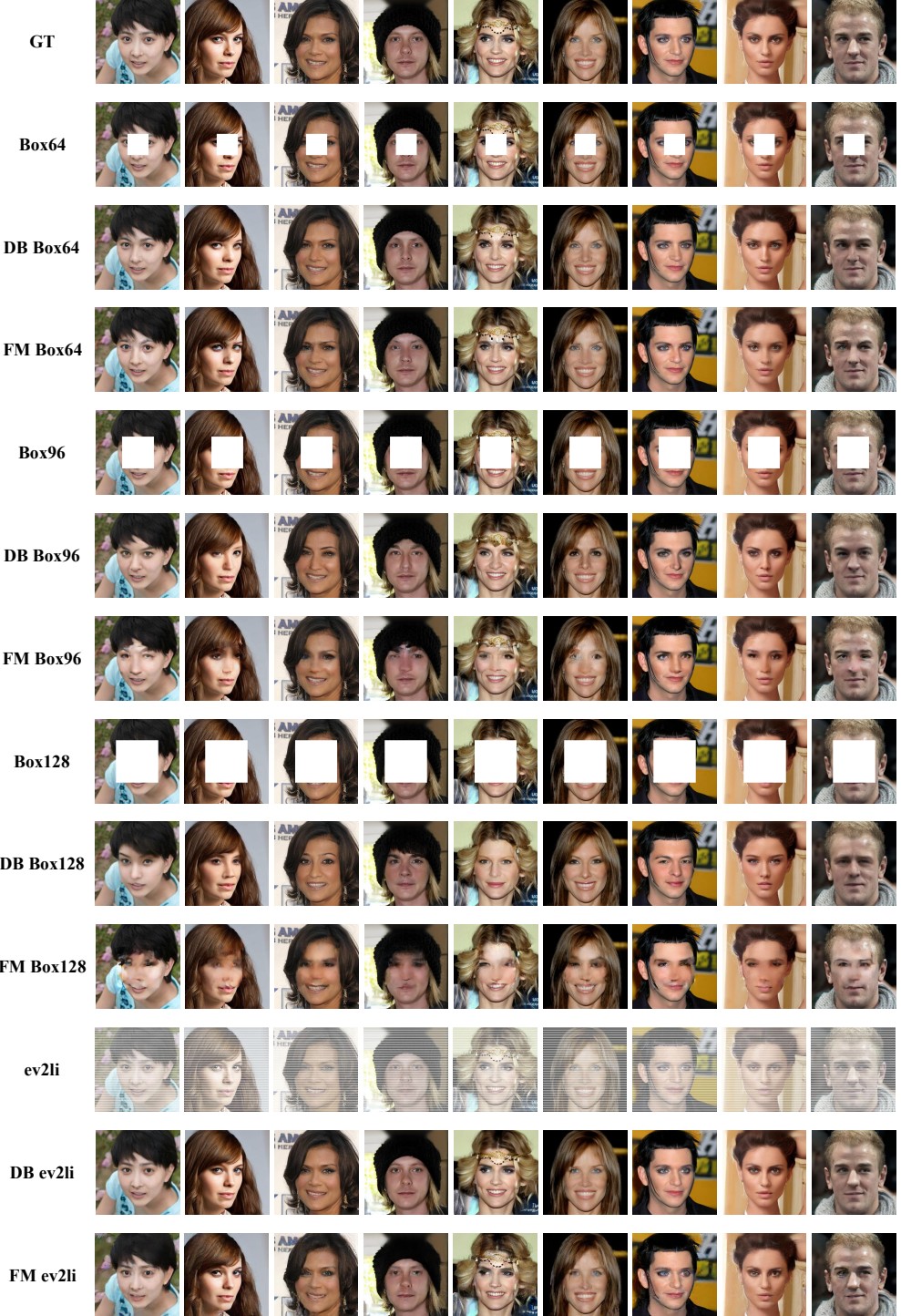

*Figure 8.* Additional Diffusion Bridge and Flow Matching visual results on Image Inpainting (different masks) with CelebA-HQ dataset.

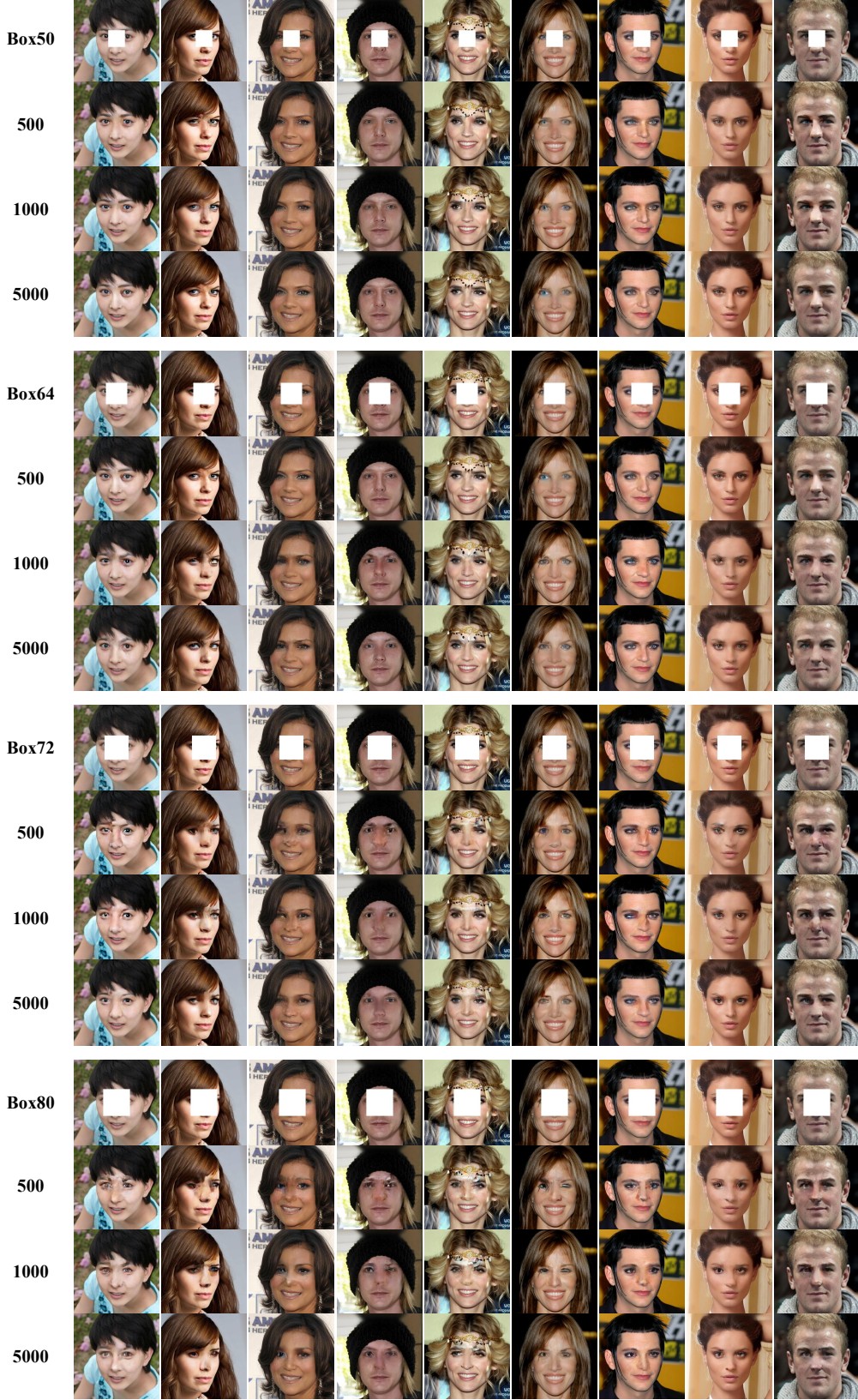

*Figure 9.* Additional Flow Matching visual results on Image Inpainting (different masks and different data size (500, 1000, 5000)) with CelebA-HQ dataset.

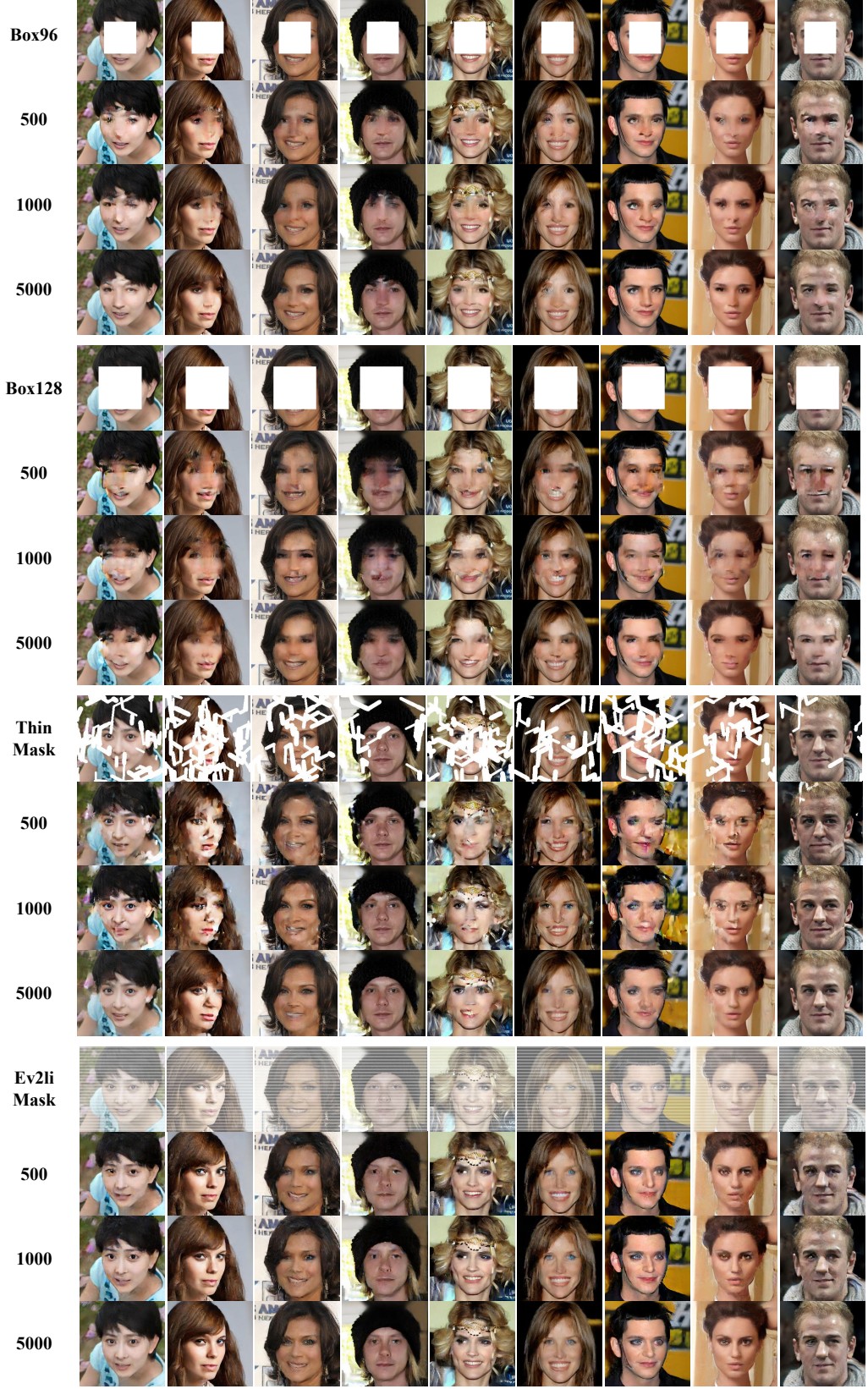

*Figure 10.* Additional Flow Matching visual results on Image Inpainting (different masks and different data size) with CelebA-HQ dataset.

