# OpenReview forum: "Diffusion Bridge or Flow Matching? A Unifying Framework and Comparative Analysis"
_ICML.cc/2026/Conference — ICML 2026 regular_

### Official Review · Reviewer_WjFU · 2026-02-25

**Soundness:** 3
**Presentation:** 3
**Significance:** 3
**Originality:** 3
**Overall Recommendation:** 4
**Confidence:** 2

**Summary:**

This paper unifies Diffusion Bridge (DB) and Flow Matching (FM) under a stochastic optimal control framework, showing that FM is a special case of DB. Theoretically, DB achieves lower optimal control cost and is more robust under limited training data, while FM degrades as data size decreases. These claims are validated through controlled experiments on CelebA-HQ across multiple image restoration tasks (inpainting, super-resolution, deblurring, denoising), varying task difficulty, dataset size, and computational cost, using the same DiT architecture in a VAE latent space for fair comparison.

**Compliance With Llm Reviewing Policy:**

Affirmed.

**Final Justification:**

The rebuttal improves the paper's clarity and i will maintain 4 (weak accept) score.

**Key Questions For Authors:**

Please see weaknesses. And I admit that I am not an expert on this topic, so I would like to see other reviewers' opinions on this topic.

**Limitations:**

yes

**Strengths And Weaknesses:**

Strengths:

1. The experimental section presents the manuscript's arguments in a detailed and precise manner, and the experiment settings are well-designed and convincing.

2. The code is publicly accessible (though the anonymous repository has expired).

Weaknesses:

Major:

1. Fig.4 demonstrates that under identical training samples, DB requires significantly more time than FM, which may explain why DB's performance slightly outperforms FM. It is recommended that the authors include a performance comparison under identical training durations for a more comprehensive analysis.

Minor:

1. Performance comparisons between DB and FM were conducted solely on the CelebA-HQ facial dataset for tasks including inpainting, deblurring, and super-resolution. It is recommended to incorporate additional natural datasets (such as DIV2K, DRealSR, etc.) to conduct a more comprehensive analysis of the performance differences between the above two paradigms.

2. In Table 1, FM sometimes achieves higher PSNR/SSIM than DB (e.g., inpainting PSNR 23.54 vs 23.57, SR SSIM 0.789 vs 0.762). The paper attributes this to DB favoring perceptual quality, but this perception-distortion trade-off deserves deeper analysis rather than a passing mention.

---

> ### Author Rebuttal · Authors · 2026-03-31
>
> Thank you for your feedback and comments.
>
> > W1: Fig.4 demonstrates that under identical training samples, DB requires significantly more time than FM, which may explain why DB's performance slightly outperforms FM. It is recommended that the authors include a performance comparison under identical training durations for a more comprehensive analysis.
>
> A: Here we provide a comparison between DB and FM under the same training budget. Taking Inpainting (Box128) task as an example, we report the FID scores of FM and DB at the same training epochs. In the early stage, DB converges more slowly and delivers inferior results than FM. However, once FM has fully converged, DB continues to improve and ultimately achieves better performance.
>
> ||Epoch=40|Epoch=60|Epoch=80|Epoch=100|Epoch=150|Epoch=180|
> |:-|:-|:-|:-|:-|:-|:-|
> ||FID↓|FID↓|FID↓|FID↓|FID↓|FID↓|
> |FM (Box128)|**19.31**|**18.32**|**18.57**|18.26|18.64|18.57|
> |DB (Box128)|25.46|25.16|24.44|**15.75**|**9.32**|**7.71**|
>
>
> > W2: Performance comparisons between DB and FM were conducted solely on the CelebA-HQ facial dataset for tasks including inpainting, deblurring, and super-resolution. It is recommended to incorporate additional natural datasets (such as DIV2K, DRealSR, etc.) to conduct a more comprehensive analysis of the performance differences between the above two paradigms.
>
> A: Our experiments are **not limited to the CelebA-HQ facial dataset**, but also include other image-to-image translation tasks on Edges2Handbags, Edges2Shoes (including diverse real-world objects), and Nights2Days (including complex natural outdoor landscapes) datasets in Section 5.2. The quantitative and qualitative results are shown in Table 1 and Figure 2.
>
> > W3: In Table 1, FM sometimes achieves higher PSNR/SSIM than DB (e.g., inpainting PSNR 23.54 vs 23.57, SR SSIM 0.789 vs 0.762). The paper attributes this to DB favoring perceptual quality, but this perception-distortion trade-off deserves deeper analysis rather than a passing mention.
>
> A: In the field of image generation/restoration, it is a common phenomenon that FM sometimes achieves higher PSNR/SSIM than DB. ODE-based models (FM) introduce zero additional noise during the sampling process, thereby generating much smoother images with better pixel-level distortion metrics (PSNR and SSIM). For example, see the results shown in Table 1–3 of GOUB [R1] and Table 1 of UniDB [R2]. During the rebuttal, we have also implemented DB with ODE sampler, which achieves higher PSNR/SSIM than DB (SDE) **(e.g., inpainting box128: PSNR 23.83 vs. 23.57, SSIM 0.750 vs. 0.741)**.
>
> Meanwhile, according to Section 4.1, DB introduced the drift term in the SDE formulation, so its trajectories are more stable and better conform to the natural data manifold. Therefore, the images generated by DB are visually more realistic and some unnatural patterns in smooth regions are avoided (better LPIPS and FID).
>
> [R1] Yue et al. "Image restoration through generalized ornstein-uhlenbeck bridge", ICML 2024.
>
> [R2] Zhu et al. "UniDB: A Unified Diffusion Bridge Framework via Stochastic Optimal Control", ICML 2025.
>
>
> > The anonymous repository has expired.
>
> A: Thank you for raising this point. We have extended its validity period to the same code repository link in the Abstract.

---

> > ### Author Rebuttal · Reviewer_WjFU · 2026-04-01
> >
> > Thank you for the rebuttal. The additional explanations are helpful.
> >
> > W2 is adequately addressed: I appreciate the clarification that the paper includes experiments beyond CelebA-HQ. W3 is partially addressed: the added explanation and references make the perception-distortion discussion more convincing, although a more systematic analysis would still be desirable. W1 is also only partially addressed, since my original question concerned comparison under the same wall-clock training time, while the rebuttal reports results at the same training epochs.
> >
> > Overall, the rebuttal improves the paper's clarity and i'll keep the positive rating, but my concern about fairness under equal training time remains only partially resolved.

---

> > > ### Author Response · Authors · 2026-04-02
> > >
> > > Thank you for acknowledging our efforts and **maintaining the positive rating!** We'd like to present further explanations on the remaining concerns.
> > >
> > > > W3 is partially addressed: the added explanation and references make the perception-distortion discussion more convincing, although a more systematic analysis would still be desirable.
> > >
> > > A: Here we provide a more systematic analysis on the perception-distortion trade-off. The advantage of ODE over SDE in PSNR/SSIM stems from determinism and invertibility. ODE establishes a bijective mapping between the input and output, preserving pixel-level and structural information with minimal deviation. Conversely, SDE injects random noise at each reverse step, which degrades fidelity to the original input. SDE excels in LPIPS and FID due to its stochasticity. The injected random noise allows the trajectory to explore the data manifold, correcting out-of-distribution inputs and generating more natural textures and diversity [R1]. Thus, deterministic ODEs favor fidelity, while stochastic SDEs favor perceptual quality.
> > >
> > > RDMD [R2] also revealed that in practice, "deterministic regression achieves high data fidelity while many unrealistic smearing artifacts are presented; Conversely, stochastic sampling can enhance perceptual quality but often at the expense of fidelity to the original image".
> > >
> > > We will add the detailed analysis in the revised version.
> > >
> > >
> > > [R1] Nie et al. "The Blessing of Randomness: SDE Beats ODE in General Diffusion-based Image Editing", ICLR 2024.
> > >
> > > [R2] Wang et al. "Reconciling Stochastic and Deterministic Strategies for Zero-shot Image Restoration using Diffusion Model in Dual", CVPR 2025.
> > >
> > >
> > > > W1 is also only partially addressed, since my original question concerned comparison under the same wall-clock training time, while the rebuttal reports results at the same training epochs. Overall, the rebuttal improves the paper's clarity and I'll keep the positive rating, but my concern about fairness under equal training time remains only partially resolved.
> > >
> > > A: We appreciate the concerns about fairness of comparison under the same wall-clock training time. **The identical training epochs can actually translate to virtually identical wall-clock training time.** Here we provide a direct runtime measurement: 60 training epochs takes **17.33 hours** for FM and **17.56 hours** for DB. The negligible difference **(less than $1.5 \%$)** confirms the equivalence between training epochs and wall-clock training time.

---

### Official Review · Reviewer_ExTc · 2026-02-28

**Soundness:** 3
**Presentation:** 3
**Significance:** 3
**Originality:** 3
**Overall Recommendation:** 4
**Confidence:** 3

**Summary:**

This paper compares the diffusion bridge and flow matching from the perspectives of Stochastic Optimal Transparent and Optimal Transport, and combines experiments to show their respective advantages and disadvantages. To be more specific, the author tends to emphasize the diffusion bridge's advantage: it is more stable and yields a smaller cost function.

**Compliance With Llm Reviewing Policy:**

Affirmed.

**Final Justification:**

The revised proof now appears reasonable to me. Based on this, I increase my score to 4.

**Key Questions For Authors:**

I list below what I believe are the questions, but I would be happy to be corrected if I misunderstood any part of the work.

1. In A.2 (the proof of Theorem 4.2), could you explain the $x_{t}^{(u)}$ in (19)? Based on my understanding, it should align with the true trajectory, which means that for DB and FM, $x_{t}^{(u)}$ should differ depending on the true trajectory. Even if this is correct and the result is “less than or equal,” you cannot claim “strictly larger” in your conclusion.
2. In A.3, the weak convergence in (23) should be based on the i.i.d. assumption or other assumptions for $x_{i}$ and $y_{i}$. Equation (24) looks wrong: if $dim_{\mathcal{H}}(\cdot) = 0$, then the $\limsup$ should also be $0$. Based on your proof, I guess the $\limsup$ should be taken directly for $\mathcal{S}_{t}^{(n)}$. I also do not understand why “the limit superior set recovers the full dimension $d$, thereby restoring absolute continuity.” How is absolute continuity derived from full dimension? For clarification, I would like you to explain in detail how (33) is derived, what assumptions you truly use, and clearly define $T_n$ in (32) and $m$ in (34). Regarding Caffarelli’s interior regularity in (30), is there a specific reference I should read? I am not familiar with this result, and I do not know whether it only requires absolute continuity or stronger density-related assumptions. Please add the reference and correct me if I am wrong.
3. Based on Appendix D, it looks like the experiments are based on $\lambda$ and $\theta_{t}$ that do not make $g_t = 1$. I understand that in the theorem, you want to keep the formula the same as flow matching. But I have two questions: first, what are the true experimental results for $g_{t}=1$, and do they align with Theorem 4.2? Second, do you have any proof that when $g_{t} \neq 1$, how the loss $\mathcal{J}(\cdot)$ changes? and wether the inequality still holds?
4. The explanation of Theorem 4.2 looks somewhat empirical; some logic is not very clear, and some concepts lack definitions. I can accept most of them, but please answer this: you claim that the “discrepancy between distributions” harms flow matching. Could you strictly define the “discrepancy between distributions” and give me some proof, or at least insight, on why using the center mask can enlarge this discrepancy between distributions?
5. Also, the code repository appears to have expired, so I cannot refer to the code at this time.

Questions 1 and 2 are central to my evaluation. A convincing clarification or resolution would positively influence my overall score.

**Limitations:**

yes

**Strengths And Weaknesses:**

## strength
1. It's a systematic comparison between the diffusion bridge and the flow matching. This paper stands out for discussing the diffusion bridge, an advanced new method for distribution-to-distribution transformation in recent years.
2. The experiments look reasonable and comprehensive, and they basically align with the theory and the main figure.
3. The writing of the main paper is clear and can capture the main idea quite quickly.

## weakness
1. Some proofs are not convincing to me and feel like they skip core steps; please see Questions 1 and 2.
2. Some experimental setups may not be well aligned with the theory; see Question 3.
3. Some induction logic in the main paper is based on heuristic intuition; see Question 4.

---

> ### Author Rebuttal · Authors · 2026-03-31
>
> We greatly appreciate your feedback and inquiries.
> > Q1: The proof of Theorem 4.2.
>
> A: You're right that $x_t^u$ in (19) should align with the true trajectory and they are different. Here we correct the proof as follows and the inequality of Theorem 4.2 still holds:
>
> Recall that both DB and FM share the same endpoints $x_0$ and $x_1$, we can replace $x_t^u$ with the interpolation combination of $x_0$ and $x_1$. Denote $\bar{\theta}_ {s:t}=\int_s^t\theta_zdz$, $\bar{\theta}_ t=\int_0^t\theta_zdz$ and $\bar{\sigma}^2_ {s:t}=\lambda^2(1-e^{-2\bar{\theta}_ {s:t}})$. According to UniDB [R1], the optimal controller $u_t^{DB}$ and the transition $x_t^u$ are
> $u_ t^{DB}=e^{-2\bar{\theta}_ {t:1}}(x_1-x_t^u)/\bar{\sigma}^2_ {t:1}$
>
> $x_t^u=\xi_tx_0+(1-\xi_t)x_1,\xi_t=e^{-\bar{\theta}_ t} \bar{\sigma}^2_ {t:1}/\bar{\sigma}^2_1$.
>
> Take $x_t^u$ into $u^{DB}_ t$, $u_ t^{DB}=e^{-2\bar{\theta}_ {t:1}}\xi_t(x_1-x_0)/\bar{\sigma}^2_ {t:1}=e^{\bar{\theta}_t}e^{-2\bar{\theta}_1}/\bar{\sigma}^2_1(x_1-x_0)$
>
> As for FM, $u_t^{FM}=x_1-x_0$ (Eqn. 11). Recall that $2\lambda^2\theta_t=g^2_t$ in UniDB, we have $g_t=1$ which implies $\theta_t=\frac{1}{2\lambda^2}$ and we also have $e^x-1\ge x$ for $x\ge0$, then
>
> $J(u_t^{DB})=0.5\int_0^1|u_t^{DB}|_2^2dt
> =0.5\int_0^1|e^{\bar{\theta}_t}e^{-2\bar{\theta}_1}/\bar{\sigma}^2_1(x_1-x_0)|_2^2dt$
>
> $=0.5|x_1-x_0|_2^2\int_0^1\frac{e^{2\bar{\theta}_t}e^{-4\bar{\theta}_1}}{\lambda^4(1 - e^{-2\bar{\theta}_1})^2}dt
> =\frac{|x_1-x_0|_2^2}{2\lambda^4 (e^{2\bar{\theta}_1}-1)^2}\int_0^1e^{t/\lambda^2}dt$
>
> $=\frac{|x_1-x_0|_2^2}{2\lambda^2(e^{1/\lambda^2}-1)^2}(e^{1/\lambda^2}-1)
> =\frac{|x_1-x_0|_2^2}{2\lambda^2(e^{1/\lambda^2}-1)}
> \le \frac{|x_1-x_0|_2^2}{2\lambda^2 (1/\lambda^2)}
> =0.5|x_1-x_0|_2^2=0.5\int_0^1|u^{FM}_t|_2^2dt=J(u_t^{FM})$ which concludes the proof.
>
> As for “strictly larger” in the conclusion, thank you for raising this point and we will remove the word “strictly”.
>
> [R1] Zhu et al. "UniDB: A Unified Diffusion Bridge Framework via Stochastic Optimal Control", ICML 2025.
> > Q2.1: i.i.d. sampling assumption.
>
> A2.1: The weak convergence here indeed relies on the assumption that $\\{(x_i,y_i)\\}_{i=1}^n$ are i.i.d. samples.
> > Q2.2: Hausdorff dimension typo.
>
> A2.2: It is a typo in Eqn. (24). The correct one is $\limsup_{n→\infty} \operatorname{dim}_{\mathcal{H}}(\mathcal{S}_t^{(n)})=d$.
> > Q2.3: Clarify full dimension and absolute continuity.
>
> A2.3: We clarify: the Hausdorff dimension recovery **is a consistency verification, not a derivation**. The limit measure $\mu_t$ is absolutely continuous since $\mu_0$ is assumed absolutely continuous, which is the prerequisite for Brenier's theorem. The full dimension $\limsup_{n→\infty} \operatorname{dim}_{\mathcal{H}}(\mathcal{S}_t^{(n)})=d$ is consistent with this property, indicating that the discrete empirical supports asymptotically fill the continuous limit support. We apologize for the misleading "thereby restoring"; it should read "consistent with."
>
> Jabin [R2] propose a novel methodology for discrete/finite data settings that **operates independently of Caffarelli's renormalization techniques**. This approach enables the attainment of approximate regularity even when the stringent assumptions of Brenier's theorem are violated (the measures fail to be absolutely continuous). **This result conceptually aligns with Remark 4.3 in our paper.**
>
> [R2] Jabin et al. "Holder and Sobolev regularity of optimal transportation potentials with rough measures", Theorem 2.5, 2.6, 2023.
> > Q2.4: Notation clarification.
>
> A2.4: $T_n$ is the optimal transport map from the empirical measure $\hat{\mu}_0^{(n)}$ to $\hat{\mu}_1^{(n)}$, and the symbol $m$ appears to be a typo.
> > Q3.1: Experimental results for $g_t=1$.
>
> A: We provide the comparison between DB with $g_t=1$ and FM in the table below and the results align with Theorem 4.2. DB with $g_t=1$ still shows better performance than FM. It confirms the observed stability is an intrinsic theoretical property of DB.
>
> ||box96||box128||
> |:-|:-|:-|:-|:-|
> ||LPIPS↓|FID↓|LPIPS↓|FID↓|
> |FM|0.060|8.18|0.106|17.84|
> |DB|**0.052**|6.25|**0.078**|**7.71**|
> |DB ($g_t=1$)|**0.052**|**5.96**|0.079|7.86|
> > Q3.2: Any proof that when $g_t\neq1$?
>
> A: When $g_t\neq1$, it's hard to prove the inequality and may require additional conditions. We're willing to explore it in future work.
> > Q4: Clarification on "Discrepancy between distributions".
>
> A: We agree that "discrepancy between distributions" here is somewhat misleading. "As the discrepancy between the terminal distributions grows" actually corresponds to Section 5.3 (Experiments under Varying Levels of Task Difficulty). As the masked center region in the inpainting task gradually increases from 50x50 to 128x128, there are more blank spaces in the image center and thus, the discrepancy between the images increases.
> > Q5: Expired code repository.
>
> A: Thanks for raising this point. We have extended its validity period to the same code repository link in Abstract.

---

> > ### Author Rebuttal · Reviewer_ExTc · 2026-04-03
> >
> > I thank the authors for the detailed rebuttal, which has clarified my concerns. The revised proof now appears reasonable to me. Based on this, I increase my score to 4.

---

> > > ### Author Response · Authors · 2026-04-04
> > >
> > > Thank you very much for your positive feedback and for increasing your score to 4. We are glad that our rebuttal and the revised proof have addressed your concerns.

---

### Official Review · Reviewer_eG6j · 2026-03-09

**Soundness:** 3
**Presentation:** 3
**Significance:** 2
**Originality:** 3
**Overall Recommendation:** 4
**Confidence:** 3

**Summary:**

This work aims to provide a unified comparison between two generative modeling paradigms, i.e., the Diffusion Bridge  (DB) and Flow Matching (FM), from the theoretical insights through the lens of Stochastic Optimal Control (SOC) and Optimal Transport (OT), as well as empirical justifications through downstream tasks such as image inpainting, deblurring, and super-resolution. The authors draw conclusions that DB offers lower cost and superior performance under limited training data, at the cost of longer training and sampling time.

**Compliance With Llm Reviewing Policy:**

Affirmed.

**Final Justification:**

Keep my positive rating after rebuttal.

**Key Questions For Authors:**

Please see my Weaknesses above.

**Limitations:**

No, the authors are encouraged to discuss the limitations of the proposed comparison framework. One potential suggestion is to discuss the real-world scenarios where the current theoretical assumptions do not hold.

**Strengths And Weaknesses:**

**Strengths:**

1.	The paper is, in general, well-written and easy to follow, with a clear positioning to tackle the research question: “Mapping one distribution to another, which is better—Diffusion Bridge or Flow Matching?”

2.	The proposed theoretical perspective to compare two modeling paradigms through SOC and OT is interesting and seems to be empirically supported through several downstream computer vision application tasks.

**Weaknesses:**

1.	While the research question aims to answer which modeling paradigms offer better distribution mapping ability, the current justification seems rather implicitly focused on the scenario where there exists a pairwise matching between two boundary distributions, which is a rather specific case that does not really hold in the generic OT problem. This is also reflected in the experimental designs, where the tasks focus more on the image translation tasks, where there exists an expected GT output for the given input. I wonder if the authors have taken the unconditional generation or more generic text-to-image generation into account, and maybe examine and compare the performance of DB and FM in those more general cases, looking into some other important criteria, such as the sample diversity, in addition to general quality.

2.	If I recall correctly, the implementation of DB for tasks like image translation typically takes both the condition and the intermediate noisy latent as input, while FM typically takes the intermediate noisy latent. Can authors elaborate on how this is handled in practice, and how this impacts the theoretical derivations?

---

> ### Author Rebuttal · Authors · 2026-03-31
>
> We sincerely thank you for your comments and feedback.
>
> > W1: While the research question aims to answer which modeling paradigms offer better distribution mapping ability, the current justification seems rather implicitly focused on the scenario where there exists a pairwise matching between two boundary distributions, which is a rather specific case that does not really hold in the generic OT problem. This is also reflected in the experimental designs, where the tasks focus more on the image translation tasks, where there exists an expected GT output for the given input. I wonder if the authors have taken the unconditional generation or more generic text-to-image generation into account, and maybe examine and compare the performance of DB and FM in those more general cases, looking into some other important criteria, such as the sample diversity, in addition to general quality.
>
> A: We appreciate your constructive suggestions. As for **unconditional generation task** (DB's one endpoint is fixed as standard Gaussian distribution), DB will reduce to diffusion models with different drift and interpolation coefficients [R1]. In Table 4 of DDBMs [R1], a performance comparison between DB and FM on unconditional generation tasks (CIFAR-10 and FFHQ-64×64) is presented, revealing marginally better performance of DB over FM. This narrower gap can attribute to that FM with linear interpolation performs better in this simpler setting (Gaussian source distribution).
>
>
> ||CIFAR-10||FFHQ-64×64||
> |:-|:-|:-|:-|:-|
> ||NFE↓|FID↓|NFE↓|FID↓|
> |Rectified Flow|127|2.58|152|4.45|
> |DDBM|35|**2.06**|79|**2.44**|
>
> As for **more generic text-to-image generation**, conducting text-to-image generation experiments for both DB and FM is highly expensive. Taking CrossFlow [R2] (a flow-based text-to-image model) as an example, it requires about **350M image-text pairs and 630 A100 GPU-days**. Given hardware/time constraints, conducting such experiments is practically infeasible during rebuttal. We appreciate the suggestions and are willing to explore them further in future work.
>
> [R1] Zhou et al. "Denoising Diffusion Bridge Models", ICLR 2024.
>
> [R2] Liu et al. "Flowing from Words to Pixels: A Noise-Free Framework for Cross-Modality Evolution", CVPR 2025.
>
> > W2: If I recall correctly, the implementation of DB for tasks like image translation typically takes both the condition and the intermediate noisy latent as input, while FM typically takes the intermediate noisy latent. Can authors elaborate on how this is handled in practice, and how this impacts the theoretical derivations?
>
> A: Our DB Latent Transformer takes $(x_t,x_T,t)$ as input, where $x_t$ and $x_T$ are combined via subtraction and concatenation operations. Specifically, we first compute $x_t-x_T$, then concatenate it with $x_T$ along the channel dimension. As for FM, it indeed typically accepts $(x_t,t)$ as model input, which is how we implemented it in this work.
>
> **The theoretical derivation itself is independent of the network structure**. In Section 4.1 and 4.2, our theoretical analysis between DB and FM focus on the forward process modelling, not the sampling process which is related to the network. **We also conducted experiments in Section 5.4 for DB vs FM with the Same Network Input to ensure fairness of this implementation.** As shown in Table 3, "although FM employs the same neural network input conditions as DB, its performance, while improved, remains inferior to that of the DB." Therefore, the performance gap results from the inherent theoretical distinctions, rather than this implementation.

---

> > ### Author Rebuttal · Reviewer_eG6j · 2026-04-03
> >
> > Thanks for the rebuttal.
> > I will maintain my positive rating. While I understand that experiments on T2I seem rather competitively expensive and too extensive during a short rebuttal period, I still believe that exploring more generic and scaled-up settings will bring a higher impact to the work, as well as the takeaway information delivered by this work.

---

> > > ### Author Response · Authors · 2026-04-03
> > >
> > > Thank you for recognition and **maintaining your positive rating**! We greatly appreciate your constructive suggestions and will actively explore these more generic and scaled-up tasks in our future work.

---

### Official Review · Reviewer_Ed75 · 2026-03-09

**Soundness:** 2
**Presentation:** 3
**Significance:** 2
**Originality:** 3
**Overall Recommendation:** 4
**Confidence:** 4

**Summary:**

This paper provides a unified theoretical and experimental comparison between Diffusion Bridge (DB) and Flow Matching (FM) frameworks for arbitrary distribution transformation. By recasting both models through the lens of Stochastic Optimal Control, the authors prove that DB possesses a lower cost function and more stable trajectories, while also demonstrating that FM’s linear interpolation coefficients lose effectiveness under small-data regimes. To validate these claims, they introduce a novel latent Transformer architecture for DB and conduct extensive benchmarks across restoration and translation tasks to delineate the respective merits of each approach.

**Compliance With Llm Reviewing Policy:**

Affirmed.

**Final Justification:**

The rebuttal addressed the concerns to some extent. Looking at the overall quality and other reviews, I believe the score can be a 4.

**Key Questions For Authors:**

1. **Cost Function Interpretation:** The authors prove that the Diffusion Bridge has a lower cost function in the Stochastic Optimal Control framework. Does this lower cost directly correlate with lower variance in the reverse-time sampling, or does it primarily affect the curvature of the ODE/SDE trajectories?
2. **Small-Data Regime:** Why exactly does the linear interpolation of Flow Matching ($t$ and $1-t$) fail when data is scarce? Is it because the model fails to learn the straight-path vector field, or is it an inherent bias in the OT formulation that requires a denser density estimation?
3. **Latent Transformer Design:** Could you elaborate on the "architectural innovations" in your DB Transformer? Specifically, how does it handle the boundary conditions of a "Bridge" (starting and ending at specific distributions) differently than a standard DiT?
4. **Trajectory Straightness:** Flow Matching is known for producing "straight" trajectories that allow for few-step sampling. Does Diffusion Bridge, despite its lower SOC cost, result in curved trajectories that require more sampling steps than FM?
5. **Hyperparameter Sensitivity:** Does DB require more careful tuning of the diffusion schedule compared to the relatively parameter-free linear schedule of Flow Matching?

**Strengths And Weaknesses:**

**Strengths**
* **Theoretical Rigor:** The paper addresses a significant gap in generative modeling by providing a unified account of DB and FM using Stochastic Optimal Control and Optimal Transport theories. Proving the lower cost function of DB offers a principled explanation for its stability.
* **Fair Benchmarking:** The authors go to great lengths to ensure a "fair comparison" by designing a new latent Transformer architecture and applying it identically to both DB and FM. This eliminates architectural bias, a common flaw in comparative studies.
* **Insightful Data-Scaling Analysis:** The observation that Flow Matching’s interpolation scheme ($\alpha_t$ and $\sigma_t$) degrades with limited training data is a highly practical contribution for researchers working with small-scale datasets.
* **Comprehensive Task Coverage:** Evaluating across Image Restoration (Inpainting, SR, Deblurring), Translation, and Style Transfer provides a holistic view of how these models perform under varying "distributional discrepancies."

**Weaknesess**
* **Architecture Novelty vs. Analysis:** While the "latent Transformer" for DB is listed as a contribution, the paper focuses more on comparison than on the architectural details. It is unclear if the performance gains come from the DB framework itself or the specific Transformer design.
* **Complexity of Stochastic Optimal Control:** The theoretical recasting is mathematically dense. The paper would benefit from a more intuitive visualization or a "plain English" summary of how a "lower cost function" translates to visual quality in specific tasks.
* **Inference Efficiency:** The paper mentions that DB has more "stable and natural trajectories," but it does not explicitly compare the sampling speed (NFE - Number of Function Evaluations) required for DB vs. FM to reach convergence.
* **Generalization to High-Resolution:** Most experiments seem focused on restoration/translation. It remains to be seen if the DB advantages hold for high-resolution text-to-image generation, where FM (like in Flux) currently dominates.

---

> ### Author Rebuttal · Authors · 2026-03-31
>
> We sincerely thank you for your comments and questions. We will provide a detailed response to these concerns.
>
> > W1: Architecture Novelty vs. Analysis.
>
> A: **The performance gains come from the DB framework itself.** Our experiments are all conducted under **the same network architecture with identical parameters** to ensure the fair comparison between FM and DB. Therefore, the performance exactly comes from the DB and FM framework thereselves.
>
>
> > W2: Complexity of Stochastic Optimal Control.
>
> A: Here we provide a more detailed explanation on how lower cost translates to visual quality. Recall that the SOC objective minimizes $\int_0^1\frac{1}{2}\|u_t\|_2^2dt$. According to UniDB [R1], a smaller cost promotes smoother, more natural diffusion paths between two fixed endpoints. A larger total cost implies **a larger controller $\|u_t\|_2^2$** and consequently, **the state $x_t$ undergoes violent fluctuations at every time step**, which may degrade visual quality during the generation process. As shown in Figure 2, taking Deblurring on the facial dataset as an example, FM with a high-cost trajectory result in disrupting local pixel continuity, causing the network to output overly smoothed, blurry patches or unnatural textures (like individual strands of hair or skin texture).
>
> [R1] Zhu et al. "UniDB: A Unified Diffusion Bridge Framework via Stochastic Optimal Control", ICML 2025.
>
>
> > W3: Inference Efficiency.
>
> A: **We did compare the convergence speeds of FM and DB across different NFE settings in Table 9 (Appendix E.3)**. The table clearly demonstrates that both FM and DB approach convergence at NFE=20, which aligns with our experimental configuration for NFE studies as detailed in the main text (Section 5.6 and Figure 4).
>
>
> > W4: Generalization to High-Resolution.
>
> A: Thank you for your constructive suggestions, but training high-resolution text-to-image generation tasks is prohibitively expensive. Taking CrossFlow [R2] (a flow-based text-to-image model) as an example, it requires **about 350M image-text pairs and 630 A100 GPU-days**. Given hardware/time constraints, conducting such experiments for both DB and FM is practically infeasible during rebuttal. We appreciate this suggestion and are willing to explore them further in future work.
>
> [R2] Liu et al. "Flowing from Words to Pixels: A Noise-Free Framework for Cross-Modality Evolution", CVPR 2025.
>
>
> > Q1: Cost Function Interpretation.
>
> A: The cost function primarily affects the curvature of the ODE/SDE trajectories. Our SOC analysis mainly focus on the **forward process** between DB and FM, which aims to transport $x_0$ to $x_1$ while minimizing the control energy $J(u_t)$. This cost function **does not directly characterize the variance of reverse-time sampling**, it describes the optimized structure of the paths between distributions and how the system completes distribution transformation under the constraint of minimizing control energy. As we mentioned in **Answer to W2**, a smaller cost promotes smoother, more natural diffusion paths between two fixed distributions. In contrast, a high-cost trajectory may result in disrupting local pixel continuity and outputting unnatural textures.
>
> > Q2: Small-Data Regime.
>
> A: It is the inherent theoretical bias. Theoretically, scarce data violates Brenier's absolute-continuity assumption (Remark 4.3). This renders McCann interpolation mathematically undefined. Such violation represents a qualitative change in measure properties, not merely a density estimation error.
>
> > Q3: Latent Transformer Design.
>
> A: Flow Matching has extensively leveraged Latent Transformers for text-to-image generation and image restoration tasks, as exemplified by CrossFlow [R2] and FM in Latent Space [R3]. In contrast, DB has yet to adopt such powerful neural architectures.
>
> Compared with the standard DiT, whose forward function takes $(x_t,t)$ as input, the forward function of our DB Transformer takes $(x_t,x_T,t)$ as input, where $x_t$ and $x_T$ are combined via subtraction and concatenation operations. Specifically, we first compute $x_t-x_T$, then concatenate it with $x_T$ along the channel dimension.
>
> [R3] Quan et al. "Flow Matching in Latent Space", Arxiv 2023.
>
> > Q4: Trajectory Straightness.
>
> A: DB does result in curved trajectories but does not require more sampling steps. We have provided the performance comparison between DB and FM across different sampling steps (NFE) in Table 9 (Appendix E.3).
>
> > Q5: Hyperparameter Sensitivity.
>
> A: No, DB does not require more careful tuning of the diffusion schedule. Our implementation adopts the default configurations from UniDB [R1]/GOUB [R4] without task-specific modifications.
>
> [R4] Yue et al. "Image restoration through generalized ornstein-uhlenbeck bridge", ICML 2024.

---

> > ### Author Rebuttal · Reviewer_Ed75 · 2026-04-05
> >
> > Thanks for the response in the rebuttal. I acknowledged the solved and unsolved issues. I changed the rating to reflect this.

---

> > > ### Author Response · Authors · 2026-04-05
> > >
> > > Thank you very much for acknowledging our efforts and raising the score to 4!

---

### Decision · Program_Chairs · 2026-04-30

**Decision:**

Accept (regular)

**Comment:**

All four reviewers gave an overall recommendation rating of Weak Accept for this submission. The paper presents a unified theoretical and experimental comparison between Diffusion Bridge and Flow Matching frameworks, using Stochastic Optimal Control and Optimal Transport to evaluate their relative merits. The reviewers appreciate that the work has theoretical rigor, fair benchmarking using a novel latent Transformer architecture, and insightful data-scaling analysis that demonstrates limitations of flow matching in small-data domains. While the reviewers initially raised concerns, the authors delivered a strong rebuttal. By supplying a revised proof and detailing wall-clock time comparisons, the authors largely resolved the core concerns. Given the solid theoretical foundation and the successful clarification of technical details during discussion, the consensus is that the paper offers well-supported insights into generative modeling paradigms. Thus the AC recommends the paper for acceptance